

# Determination of pressure baseline corrections for clumped-isotope signals with complex peak shapes

Stephan Räss [1,2], Peter Nyfeler [1,2], Paul Wheeler [3], Will Price [3], and Markus Christian Leuenberger [1,2]

[1]Climate and Environmental Physics, University of Bern, 3012 Bern, Switzerland
[2]Oeschger Centre for Climate Change Research, University of Bern, 3012 Bern, Switzerland
[3]Elementar UK Ltd., Isoprime House, Earl Road, Cheadle Hulme, Stockport, SK8 6PT, United Kingdom

**Correspondence:** Stephan Räss (stephan.raess@unibe.ch)

**Abstract.**

Pressure baseline corrections have been proposed to mitigate pressure-dependent background effects and reduce the apparent dependence of $\Delta_{47}$ on $\delta_{47}$ (non-linearity) observed in clumped-isotope studies of $CO_2$. In this work, we describe the determination of pressure baseline corrections for signals whose peak tops vary considerably across their width. Our study focuses on peaks with very small signal-to-baseline ratios (1.005 to 1.025) generated by the clumped isotopes $^{17}O^{18}O$ (linearly increasing peak top) and $^{18}O^{18}O$ (negatively curved peak top). The measurements were all performed in pure-oxygen gas using the compact, low-mass-resolution Elementar isoprime precisION Isotope Ratio Mass Spectrometer. We demonstrate that our corrections significantly reduce the influence of secondary electrons and that the adjusted clumped-isotope signals correctly increase with signal intensity. Furthermore, we extensively discuss correction procedures of varying complexity and explain why the best results were obtained by predicting multiple background values from the corresponding on-peak signals. Through this approach, we typically achieved standard deviations around $1 \cdot 10^{-9}$ (35/32), 0.2 ‰ ($\delta_{35}$), 0.5 ‰ ($\Delta_{35}$), $7 \cdot 10^{-9}$ (36/32), 0.1 ‰ ($\delta_{36}$) and 0.1 ‰ ($\Delta_{36}$) for at least 120 intervals (20 s integration). For the capital delta values, this corresponds to standard errors of the mean of less than 0.05 ‰, achieved with a total integration and analysis time of approximately 40 min and 6 h, respectively. We also show that the uncertainties of certain measurement parameters can be further reduced by optimising the measurement position (acceleration voltage) and applying additional drift corrections. For instance, for 35/32- and 36/32-related parameters we observed improvements of up to 1 order of magnitude and a factor of 7, respectively. Based on Monte Carlo simulations, we also show that the main uncertainties in our capital delta values are related to the on-peak signals, predicted backgrounds and the peak top curvature (only for $\Delta_{36}$). Additionally, we present a brief study on the influence of pressure baseline corrections on major oxygen-isotope ratios and their delta values. While these corrections had an insignificant effect on their uncertainties, the absolute values of 33/32 and 34/32 changed markedly.

## 1 Introduction

While characterising our Elementar isoprime precisION Isotope Ratio Mass Spectrometer (IRMS), we observed that our device is sensitive enough to measure the multiply-substituted oxygen isotopologues $^{17}O^{18}O$ and $^{18}O^{18}O$ in pure-oxygen gas, despite its low mass resolution and its use of $10^{11}$ $\Omega$ resistors on the corresponding cups. We found that the peak shapes of these





multiply-substituted isotopologues (clumped isotopes) are significantly influenced by the pressure baseline (PBL) (Räss, 2023). Typically, this effect is most pronounced for $^{18}O^{18}O$, whose peak top (PT) is usually negatively curved (see Sect. 3.1). Please note that a clumped isotope is an isotopologue containing at least two rare isotopes and that the pressure baseline denotes the baseline signal recorded in the presence of gas (He, 2012).

Commonly, background (BG) signals are determined by shutting off the gas supply to the IRMS and recording the signals
under the current tuning conditions (i.e., with all beams properly focused into cups). The corresponding averages, which we refer to as „collector zeros", are then subtracted from the measurement signals (He, 2012; Elementar, 2017). Subtracting collector zeros from our small clumped-isotope signals resulted in negative values, though, indicating that these values are not an appropriate representation of the background (Räss, 2023). Bernasconi (2013) concluded that this effect mainly results from secondary electrons; as more gas is admitted to the mass spectrometer, more secondary electrons are produced, causing
the measurement signals to shift to negative values. Additionally, the amount of secondary electrons depends on the source tuning parameters and collector arrangement (Fiebig, 2016). Beyond secondary electrons, He (2012) also mentioned that broadening (tailing) of dominant ion beams can affect the baseline signal. Broadening, which depends on ion beam intensity and focusing, is caused by Coulombic repulsion and scattering (He, 2012).

To determine accurate background values, He (2012), who measure clumped-isotopes of $CO_2$, suggested so-called „pressure
baseline corrections". One proposed method involves performing so-called „PBL cycles" before and after the acquisition of on-peak gas cycles, during which background values are recorded for each collector. For this purpose, the acceleration voltage (AV) is varied to record values in a background region that is adjacent to the peak. The PBL readings obtained before and after the on-peak gas cycles are in turn interpolated to determine the PBL values for each cycle, which are subtracted from the corresponding raw signals. A second method presented by He (2012), involves monitoring the signal of the mass-to-charge
ratio $(m/z)$ 49 during the on-peak gas cycles, slightly correcting this value to deduce its PBL and scaling this signal to estimate the PBL of $m/z = 47\ \mathrm{u\,e^{-1}}$. This method can be applied during on-peak gas cycles and is justified because baselines of $CO_2$ components hardly change during an acquisition (He, 2012). It is also worth mentioning that they selected the $m/z = 49\ \mathrm{u\,e^{-1}}$ cup as baseline tracker due to its high signal-to-baseline ratio, its wide size and sensitivity (He, 2012). To scale the signals they use linear regressions inferred from correlations between different signals (He, 2012). Additionally, He (2012) apply trend
corrections determined by monitoring baseline-to-baseline ratios of $CO_2$ collectors, although this trend is relatively small.

For the determination of PBL corrections, Bernasconi (2013) suggested performing acceleration voltage scans around the peaks at different partial pressures of $CO_2$. These scans can then be used to determine the relationships between the on-peak signals and the minimum background value of the beams (left or right of the corresponding peaks) (Bernasconi, 2013). They also suggest inferring the $m/z = 47\ \mathrm{u\,e^{-1}}$ background directly from the measured $m/z = 49\ \mathrm{u\,e^{-1}}$ on-peak signal (or using
the corresponding relationship), which is similar to one of the methods proposed by He (2012). As mentioned earlier, the former method is only feasible if the cups are sufficiently wide.

The aforementioned pressure baseline corrections were developed to address the observed drifts in the slopes of linear regression lines for heated gas (HG) and equilibrated gas (EG) corrections, which were proposed by Huntington (2009) and



Dennis (2011), respectively. Essentially, the purpose of these corrections is to mitigate the apparent dependence of $\Delta_{47}$ on $\delta_{47}$ (non-linearity), account for scale compression and enable inter-laboratory comparisons by calibrating/standardising the measured $\Delta_{47}$ values (He, 2012; Huntington, 2023). The origin of non-linearity as well as the development of these corrections was summarised by Bernasconi (2018), who also report an alternative approach for inter-laboratory data comparability of clumped-isotope measurements of $CO_2$ based on carbonate standards.


The most prominent quantity of clumped-isotope studies is normally the capital delta value, which is generally defined as

$$\Delta_A(‰) = \frac{{}^A R - {}^A R^*}{{}^A R^*} \cdot 1000\ ‰ = \left[ \frac{{}^A R}{{}^A R^*} - 1 \right] \cdot 1000\ ‰. \tag{1}$$

In Eq. (1), $A$ denotes the cardinal mass of the major isotopologue, $R$ is the isotope ratio measured in the sample and $R^*$ is the expected value for the same sample if all isotopes were stochastically distributed (Wang, 2004; Huntington, 2023). It is
worth noting that $\Delta_{47}$ often refers to the temperature-dependent mass-47 anomaly, which is a modified version of Eq. (1) and relates the clumped-isotope composition to temperature (Huntington, 2023) (their supplement). The capital delta value given in Eq. (1) has a similar structure as the bulk isotopic composition known from conventional stable-isotope studies, which is normally expressed as

$$\delta_A(‰) = \left[ \frac{{}^A R_{SA}}{{}^A R_{ST}} - 1 \right] \cdot 1000\ ‰. \tag{2}$$

In Eq. (2), ${}^A R_{SA}$ and ${}^A R_{ST}$ denote the isotope ratios measured in the sample (SA) and standard (ST) gas, respectively. Despite the structural similarity of the two delta values, their meanings are quite different. The capital delta value compares the measured abundance of a multiply-substituted isotopologue to the abundance of the same isotopologue if the isotopes in the sample conformed to a stochastic (random) distribution (Eiler, 2007). By construction, capital delta values are independent of the analyte concentration (Huntington, 2023) (their supplement). Nonetheless, effects like non-linearity must be considered,
as they can lead to deviations from the expected results. Although HG and EG corrections are common procedures for addressing such problems, they have various disadvantages: they introduce additional uncertainties, require monitoring to detect significant deviations and are time-consuming because they involve measuring gases equilibrated at different temperatures (He, 2012). Furthermore, these corrections have to be updated on a time-scale of weeks due to drifts (He, 2012). However, He (2012) demonstrated a strong correlation between temporal drifts of the pressure baseline and drifts in the apparent relationship be-
tween delta and capital delta values, suggesting that this relationship primarily results from pressure baseline effects. He (2012) showed that PBL corrections can reduce the dependence of the measured capital delta value on the corresponding delta value by up to 1 order of magnitude, diminish errors associated with EG corrections and improve the stability of the system, as well as the predictive power of HG lines. The PBL approach is so effective that He (2012) achieved external precisions of $\Delta_{47}$ close to the instrumental uncertainty.






The observations by He (2012) underscore the importance of adequate PBL corrections, without which high-precision measurements of clumped-isotopes appear challenging. To correct our measurements of clumped isotopes of oxygen, we attempted to determine pressure baseline corrections following the method presented by Bernasconi (2013) (and He (2012)). However, due to the complex shapes of our clumped-isotope peaks and suboptimal correlations between on-peak signals and background values, these corrections required adaptation (see Sect. 4).

To the best of our knowledge, pressure baseline corrections for non-square-shaped peaks such as ours are novel. In addition to determining these corrections and evaluating their impact on isotope ratios, delta values and capital delta values ($\Delta_{35}$ and $\Delta_{36}$) measured in pure-oxygen gas, we present analyses of the background evolution over time. Moreover, we report the effects of the corresponding corrections on the measurement precision and discuss the influence of PBL corrections on major oxygen signals. Our article concludes with a discussion on the impact of the measurement position (acceleration voltage at which measurements are performed) on isotope ratios and delta values. It should be noted that our studies primarily focus on precision rather than accuracy due to the lack of a proper absolute calibration.

## 2   Measurement setup

When gas is admitted to an IRMS, the molecules are typically ionised through electron impact ionisation in the device's source. After ionisation, the ions are accelerated in an electric field and then exposed to a magnetic field, where they experience the Lorentz force. As a result, the ions are deflected and grouped into so-called „ion beams" according to their mass-to-charge ratio. The higher the mass resolution of the mass spectrometer, the greater the separation of ions with different mass-to-charge ratios (i.e., better-defined ion beams). Eventually, the beams hit spatially separated Faraday cups, where the electric currents generated by the different ion beams are measured.

We use an Elementar isoprime precisION IRMS, which has a mass resolution of approximately $110\ \mathrm{m}\,\Delta\mathrm{m}^{-1}$ at $10\,\%$ valley separation and can detected masses up to $96\ \mathrm{u}$ (Elementar, 2022). The installed Faraday cup array consists of 10 cups and is designed to measure the mass-to-charge ratios 28 to 30, 32 to 36, 40 and 44 in air components ($N_2$, $O_2$, Ar and $CO_2$). Each cup can use one of two resistors. The resistances of the low and high gain resistors are $10^9\ \Omega$ and $10^{11}\ \Omega$, respectively. The low gain resistor is typically used for the mass-to-charge ratios 28, 32, 40 and 44. The maximum detectable signal is just below $100\ \mathrm{V}$, which corresponds to currents around $1 \cdot 10^{-7}\ \mathrm{A}$ (low gain) and $1 \cdot 10^{-9}\ \mathrm{A}$ (high gain). The software for communicating with our instrument is IonOS (version 4.5), which calculates delta values as documented in the appendix of Räss (2023).

Since the mass separation is based on the Lorentz force, it depends on the magnetic field, the ionic charge and the velocity of the ions. The ions' velocities, in turn, depend on the electric field (acceleration voltage) that is applied. Therefore, to obtain meaningful measurement results, the acceleration voltage (AV), the magnetic field, the positions of the Faraday cups and the cup sizes must be chosen appropriately. Nonetheless, mass spectrometers are dynamic systems and the optimal settings vary over time. A common procedure to account for this issue involves scanning the analyte over a range of acceleration voltages, analysing the peak spectrum and then readjusting the acceleration voltage if necessary. Fortunately, most modern mass spectrometers can autonomously determine a favourable acceleration voltage.





To collect as many ions as possible, all ion beams of interest must hit the centres of the corresponding cups, which requires an ideal geometrical alignment of the cups. However, the smaller the mass resolution of the mass spectrometer, the more challenging it becomes to achieve this goal.

In principle, the measurement signal increases with the number of ions collected by a Faraday cup. Admitting more gas to the mass spectrometer may also increase the amount of secondary electrons, though, which can significantly reduce the measurement signal if they are collected (see Fig. 1). Additionally, it is worth noting that even with the admission valve closed and the source turned-off, non-zero measurement signals are obtained, primarily due to electronic noise and residual gas. The latter contribution is mainly due to insufficient evacuation and adsorption/desorption effects. Hence, when evaluating measurement signals, noise, secondary electrons and residual gas must be considered. We refer to the combination of these contributions as „background".

Clumped-isotope signals are usually small. Thus, accurately determining the background and measurement position is crucial for obtaining reliable results. In the following sections, we demonstrate how these parameters can be determined and optimised for measurements of pure-oxygen gas ($m/z = 32\ \mathrm{u\,e^{-1}}$ to $m/z = 36\ \mathrm{u\,e^{-1}}$). However, the general principles apply to any detectable mass component. For our measurements, we used gas from three different steel cylinders, denoted as SC 84567 ($O_2 \geq 99.998\ \%$), SC 62349 ($O_2 \geq 99.9995\ \%$) and SC 540546 ($O_2 \geq 99.9995\ \%$). Gas was admitted to our IRMS through a custom-built open-split-based dual-inlet system, referred to as NIS-II (New Inlet System II) (Räss, 2023), as well as through the conventional changeover-valve-based Elementar iso DUAL INLET. Measurements were primarily performed at an acceleration voltage around 4455 V, which is close to the centre of the $m/z = 35\ \mathrm{u\,e^{-1}}$ peak. For certain experiments we also measured around 4450 V (close to the left edges of the $m/z = 33\ \mathrm{u\,e^{-1}}$ and $m/z = 34\ \mathrm{u\,e^{-1}}$ peaks) and 4465 V (close to the right edge of the $m/z = 36\ \mathrm{u\,e^{-1}}$ peak).

Hereafter, we use the term „AV scan" for a measurement in which the analyte is scanned over a range of acceleration voltages. The variable we use to denote the AV is $U_{av}$. For simplicity, we omit the units of mass-to-charge ratios (e.g., $m/z = 32$ instead of $m/z = 32\ \mathrm{u\,e^{-1}}$). Moreover, in the subscript of our delta and capital delta values, only the minor mass component is indicated (e.g., $\delta_{35}$ for the delta value referring to the isotope ratio 35/32). All our oxygen isotope ratios were determined with respect to $m/z = 16$ or $m/z = 32$.

## 3 Background

One of the main challenges in clumped-isotope measurements is precisely determining the background. Typically, clumped-isotope signals are so small that the signal-to-baseline ratios are close to 1. For the mass-to-charge ratios $m/z = 35$ and $m/z = 36$ measured in pure-oxygen gas (cylinder SC 632349, $m/z = 32$ signal around $3.3 \cdot 10^{-8}$ A and $U_{av} = 4455$ V), the signal-to-baseline ratios are approximately 1.005 and 1.025, respectively. In contrast, the signal-to-baseline ratios for $m/z = 32$, $m/z = 33$ and $m/z = 34$ are about 32.605, 3.641 and 25.503, respectively. For the baseline values, we used off-peak signals that were determined as described in Sect. 4. Using collector zero values to estimate the baseline, we obtained





signal-to-baseline ratios of 32.610 for $m/z = 32$, 3.355 for $m/z = 33$, 13.620 for $m/z = 34$, 0.919 for $m/z = 35$ and 0.698 for $m/z = 36$. Since we observed distinct peaks for $m/z = 35$ and $m/z = 36$, but their signal-to-baseline ratios are less than one, these values suggest that the collector zero values do not accurately represent the background of the clumped isotope signals.

In the following subsections, we first focus on the composition of the background and then on its stability. All measurements were performed using pure-oxygen gas. Our target clumped isotopes are $^{17}O^{18}O$ ($m/z = 35$) and $^{18}O^{18}O$ ($m/z = 36$). Due to isobaric interferences, with our setup, it is not possible to distinguish the clumped isotope $^{17}O^{17}O$ ($m/z = 34$) from $^{16}O^{18}O$.

## 3.1 Background composition

As stated previously, the background is influenced by electronic noise, secondary electrons and residual gas. While reducing electronic noise is challenging, most residual gas can be eliminated by thoroughly evacuating the mass spectrometer. Nevertheless, adsorption/desorption effects on metal surfaces can still lead to analyte contamination later on (Leuenberger, 2015). Regarding secondary electrons, the amount of gas plays a major role; the more gas is admitted, the greater the number of ions produced and the higher the emission of secondary electrons (see Fig. 1). Due to their negative charge, secondary electrons reduce both the measurement signal and the background. For dominant peaks, admitting more gas to the IRMS typically increases the signal. For off-peak and minor signals, such as $m/z = 36$, the net effect can be negative, though. The comparison of $m/z = 36$ with $m/z = 33$ and $m/z = 34$ peaks in Fig. 1 clearly demonstrates this effect.

Electron suppressors installed at the top of our Faraday cups can partially reduce the amount of secondary electrons. Another approach is to adjust the mass spectrometer's tuning to minimise the difference between the PBL recorded with and without sample gas (He, 2012). In Fig. 2, we show a series of AV scans performed at different electron suppressor voltages. It can be observed that more negative voltages result in less negative signals because at lower voltages more electrons are repelled. However, our measurements indicate that $m/z = 36$ signals saturate at electron suppressor voltages around -100 V; the factory default is typically around -38 V (Elementar, 2017). Most importantly, Fig. 2 illustrates that applying a negative potential to the Faraday cups is insufficient to achieve $m/z = 36$ signals that are higher than the collector zero value, resulting in negative background-corrected signals. This further indicates that collector zero values may not always accurately represent the background and that the background should be determined with the admission valve open.

In contrast, for $m/z = 35$ measurements, the signal saturates at electron suppressor voltages around -140 V. Therefore, we typically apply approximately -140 V instead of -100 V for pure-oxygen gas measurements. Additionally, our data suggest that between -20 V and -140 V, the relative signal increase for $m/z = 35$ is about 50 % less compared to $m/z = 36$. Specifically, at -20 V, the $m/z = 35$ signal is $-4.6 \cdot 10^{-13}$ A and at -140 V, it is $-2.2 \cdot 10^{-13}$ A.

According to He (2012), the broadening of dominant peaks may also influence the background of adjacent clumped-isotope peaks. Although we observed a positive correlation between the peak width and signal intensity for dominant oxygen signals



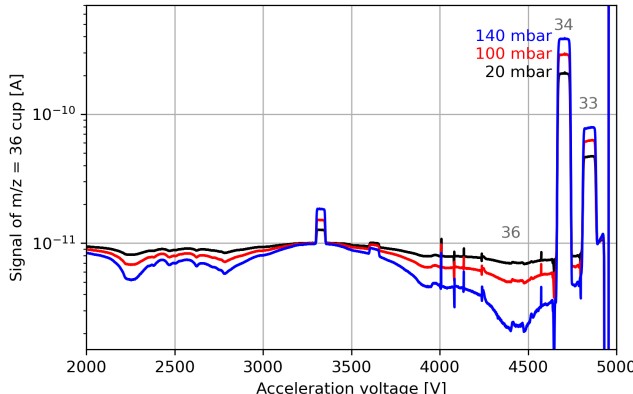

**Figure 1.** Uncorrected signals of the $m/z = 36$ cup recorded during acceleration voltage scans of pure-oxygen gas (cylinder SC 62349). The scans were performed at three different pressures of the NIS-II container (20 mbar, 100 mbar and 140 mbar), resulting in different signal intensities; the corresponding $m/z = 32$ signal intensities were approximately $4.7 \cdot 10^{-8}$ A, $6.8 \cdot 10^{-8}$ A and $9.2 \cdot 10^{-8}$ A, respectively (measured at acceleration voltages around 4455 V).

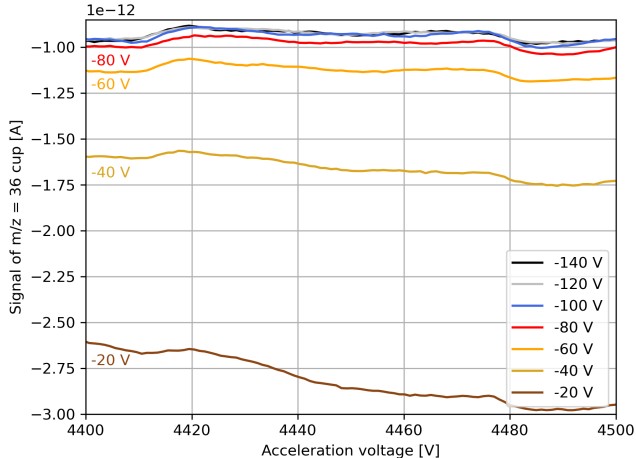

**Figure 2.** Signals of the $m/z = 36$ cup recorded during acceleration voltage scans of pure-oxygen gas (cylinder SC 84567). The scans were conducted at various electron suppressor voltages ranging from -20 V to -140 V. The signals were corrected using the collector zero value of the $m/z = 36$ cup, which was approximately $1.005 \cdot 10^{-11}$ A.

($m/z = 32$ to $m/z = 34$), this effect is too minor to significantly impact the $m/z = 35$ and $m/z = 36$ peaks. A comparison of an AV scan of pure-oxygen gas conducted at an $m/z = 32$ signal intensity of approximately $9 \cdot 10^{-8}$ A versus one performed at $1 \cdot 10^{-8}$ A revealed that, in terms of the acceleration voltage, the width of the $m/z = 34$ peak changes by less than 20 V. This change is roughly an order of magnitude smaller than the distance between $m/z = 34$ and $m/z = 35$ peaks (distance determined using the $m/z = 34$ cup). Nevertheless, we also observed that reducing the $m/z = 34$ signal can lead to an increase






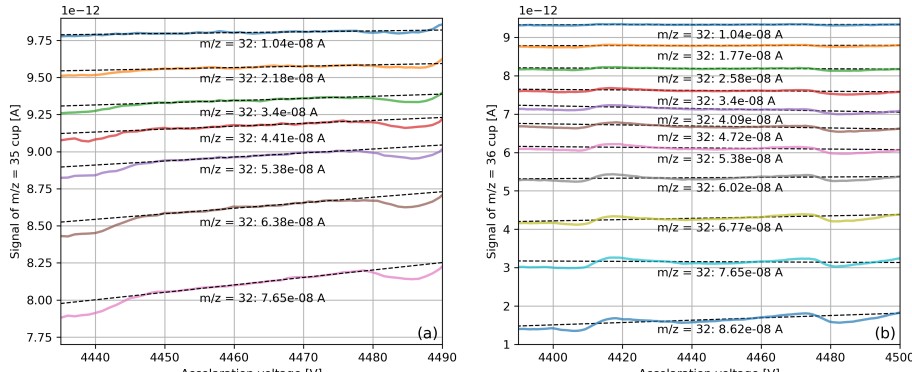

**Figure 3.** Uncorrected signals of the (a) $m/z = 35$ and (b) $m/z = 36$ cups around their peaks, with linear regressions fitted to the peak tops (4450 V to 4475 V for $m/z = 35$ and 4420 V to 4470 V for $m/z = 36$). The signals were recorded during acceleration voltage scans of pure-oxygen gas (cylinder SC 62349) performed at different trap currents. The corresponding $m/z = 32$ signal intensities (evaluated at 4455 V) are indicated below each peak.

in the background around the peak. In contrast to peak broadening, this effect can notably influence the background of our $m/z = 35$ signal.

### 3.2 Background stability

Mass spectrometers are dynamic systems whose backgrounds are subject to change. Thus, for high-precision measurements, it
is vital to monitor and assess the robustness of the background. In this subsection, we present various measurements of pure-oxygen gas performed with our Elementar isoprime precisION. Based on these measurements, we show how the backgrounds of the $m/z = 35$ and $m/z = 36$ signals vary with signal intensity and evolve over time.

To study the background of clumped oxygen isotopes in detail, we conducted AV scans with the NIS-II and varied the signal
intensity through the trap current (TC); higher trap currents result in more electrons being emitted from the filament, which in turn leads to a higher number of ions. In Fig. 3, the peaks of the recorded $m/z = 35$ and $m/z = 36$ signals are shown, along with the corresponding $m/z = 32$ signal intensities.

We also estimated the slopes of the peak tops for $m/z = 35$ (4450 V to 4475 V) and $m/z = 36$ (4420 V to 4470 V) signals using linear regression. In Fig. 4, we illustrate how these slopes change as a function of the $m/z = 32$ signal. The same figure
includes data from a second measurement series in which the reference bellow of the Elementar iso DUAL INLET was filled. When gas is continuously admitted to the mass spectrometer, the signal gradually decreases (see $m/z = 32$ signal in Fig. 5). Thus, performing AV scans periodically allows for the collection of spectra at different signal intensities. We refer to such measurements as „pressure-decrease measurements".

Most importantly, the aforementioned figures illustrate that signal variations affect not only the magnitude of the peaks but
also their shape – otherwise the slopes of the $m/z = 35$ and $m/z = 36$ peaks would not change. Since these peaks are close



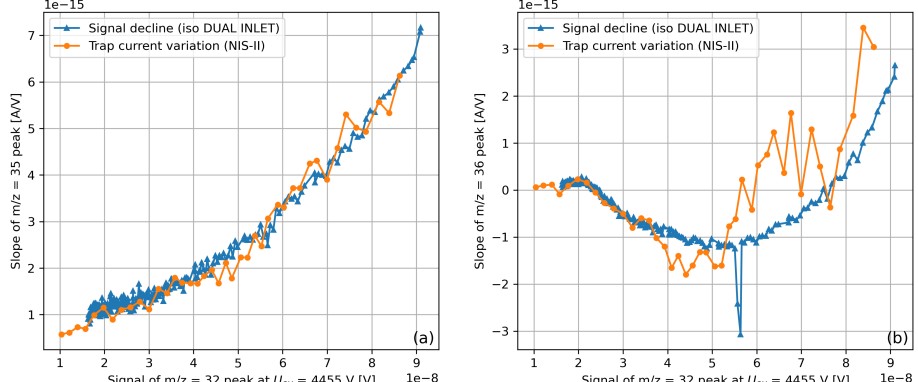

**Figure 4.** Slopes of linear peak top fits as a function of the signal intensity, computed as shown in Fig. 3. Panel (a) presents the data for the $m/z = 35$ peak tops (4450 V to 4475 V) and panel (b) shows those for $m/z = 36$ (4420 V to 4470 V). The corresponding data were obtained from a series of acceleration voltage scans of pure-oxygen gases performed at different signal intensities. For one series, gas was admitted to the mass spectrometer using the NIS-II and the signal intensity was varied by altering the trap current (see Fig. 3). For the second series, pure-oxygen gas was filled into the reference bellow of the iso DUAL INLET and then acceleration voltage scans were performed while the signal intensity decreased due to the steady consumption of gas. For NIS-II (trap current variation) and iso DUAL INLET (pressure decline) measurements, gas was sourced from the cylinders SC 632349 and SC 540546, respectively.

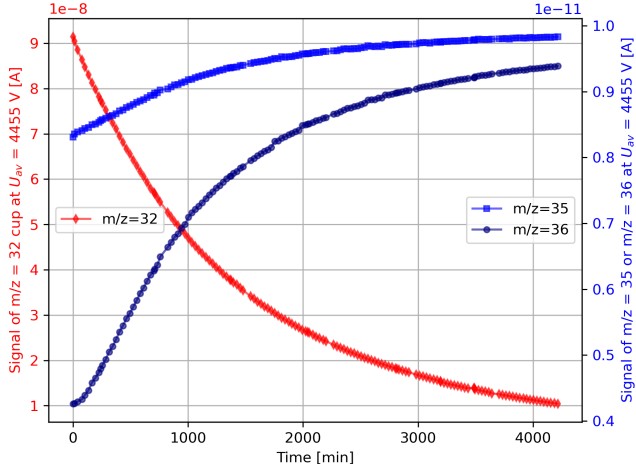

**Figure 5.** Uncorrected $m/z = 32$ (left y-axis), $m/z = 35$ and $m/z = 36$ signals (right y-axis) measured at 4455 V on the corresponding cups. The values were extracted from a series of acceleration voltage scans of pure-oxygen gas (cylinder SC 540546) performed at different signal intensities. The gas was filled into the reference bellow of the iso DUAL INLET and due to the steady gas consumption, the signal gradually decreased. The x-axis indicates the start time of each scan relative to the first measurement.

to their backgrounds, variations in the background have a substantial impact. For example, the off-peak signals to the right of the $m/z = 36$ peak increase with the $m/z = 32$ signal intensity, while those to the left decrease (see Fig. 3). Consequently,





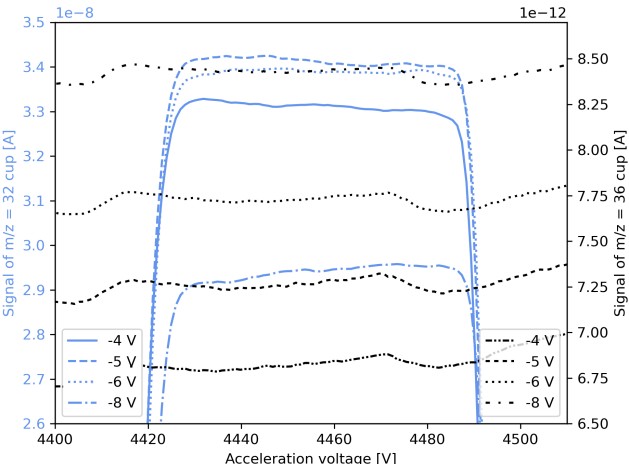

**Figure 6.** Uncorrected signals of the $m/z = 32$ and $m/z = 36$ cups recorded during acceleration voltage scans of pure-oxygen gas (cylinder SC 62349). The measurements were performed with the Elementar isoprime precisION and the NIS-II at ion repeller voltages between -4 V and -8 V.

the slope of the $m/z = 36$ peak increases as a function of the $m/z = 32$ signal. This effect makes it difficult to accurately predict the shapes of $m/z = 35$ and $m/z = 36$ peaks at a given signal intensity based solely on an AV scan performed at a

different intensity. We recommend conducting AV scans at various signal intensities instead. Two common methods for quickly varying the intensity are altering the TC and compressing the bellow (for the iso DUAL INLET only). Using the NIS-II, signal variation can also be achieved by altering the container pressure. This method is more time-consuming than the other two, though. Another noteworthy observation from Fig. 4 is that the shapes of the $m/z = 35$ and $m/z = 36$ peaks do not vary identically. Generally speaking, the variation in peak shape appears to be similar for both inlet systems and the method used to

vary the signal intensity, though (see Fig. 4).

The Elementar isoprime precisION allows adjustment of peak top tilt by varying the ion repeller (IR) voltage. However, if not all of the peaks exhibit the same trend (i.e., all increasing or decreasing with AV), it is not possible to flatten all peak tops with a single IR setting. Furthermore, we observed that variations of the IR voltage have little impact on the shapes of the $m/z = 35$ and $m/z = 36$ peaks (see Fig. 6).


The most drastic changes in background signals were typically observed after filament exchanges, which might also be related to the re-tuning of the mass spectrometer that involves adjustments to the acceleration voltage, Z-plate voltage and half plate differential voltage. After such exchanges, the slopes and intensities of the clumped-isotope signals at comparable $m/z = 32$ signal intensities can change markedly. For example, after the filament exchange in December, 2022, at an acceleration

voltage of 4455 V, the uncorrected (raw) $m/z = 36$ signal was approximately $2.70 \cdot 10^{-12}$ A at an $m/z = 32$ signal of $5.91 \cdot 10^{-8}$ A. In contrast, after the filament exchange in September, 2023, the raw $m/z = 36$ signal was about $5.57 \cdot 10^{-12}$ A at an $m/z = 32$ signal of $5.99 \cdot 10^{-8}$ A (see Fig. 7). It is striking that the $m/z = 36$ signal intensities, as well as the slopes of the




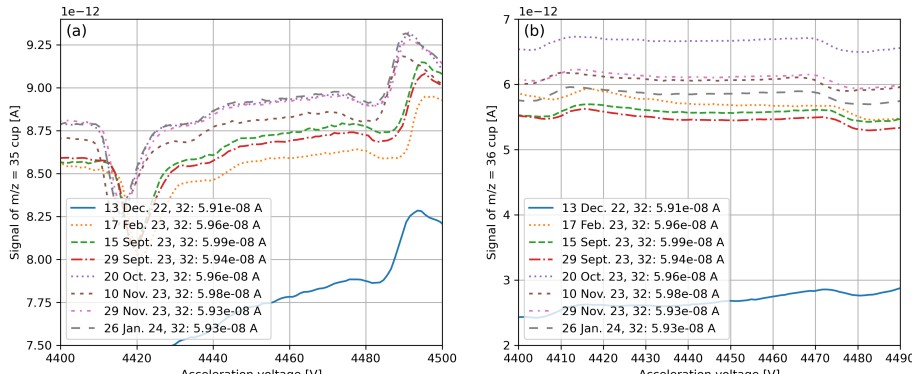

**Figure 7.** Uncorrected (a) $m/z = 35$ and (b) $m/z = 36$ peaks recorded during acceleration voltage scans of pure-oxygen gas (cylinder SC 540546) performed on different days (see legend). The gas was admitted through the iso DUAL INLET and all scans were recorded at $m/z = 32$ signal intensities around $6 \cdot 10^{-8}$ A (in the legend denoted as „32"). The mass spectrometer's filament was exchanged on December 27, 2022, February 10, 2023, September 13, 2023, and October 16, 2023.

peak tops, turned out to be significantly different at a comparable $m/z = 32$ intensity. Additionally, the slope of the $m/z = 35$ peak varied, though less markedly than that of the $m/z = 36$ peak. The aforementioned changes are most clearly illustrated by

the data presented in Fig. 8.

Notably, the signals recorded on October 20, 2023, and November 10, 2023, show differences in the left side of the $m/z = 36$ peak top and the background left of the peak, despite no filament exchange between these dates (see Fig. 8). The subsequent measurement series on November 29, 2023, closely resembles the previous series, though.

Regarding $m/z = 35$ signals, measurements carried out between October 20, 2023, and January 26, 2024, are comparable.

Moreover, Fig. 8 shows that, in terms of the acceleration voltage, the position of the $m/z = 35$ and $m/z = 36$ peaks varied by 2 V to 8 V during this period.

As background signals evolve over time, they ought to be monitored regularly. This is commonly done by measuring a standard gas with a known isotopic composition; if a major discrepancy is detected, the background correction must be adjusted. In the following, we discuss how oxygen measurements are influenced by the choice of the measurement position and the type

of background correction.

## 4    Pressure baseline corrections

As explained in previous sections, it is common practice to correct raw IRMS data by subtracting the collector zeros from the uncorrected measurement signals. Nevertheless, we have observed that the baselines of corrected signals can still be offset from zero. Although recording the collector zeros immediately before a measurement allows for the assessment of the current

state of the mass spectrometer (including noise and residual gas), the pressure-dependent non-linearity induced by secondary electrons cannot be accounted for when the admission valve is closed. Small signals are particularly affected by inappropriate





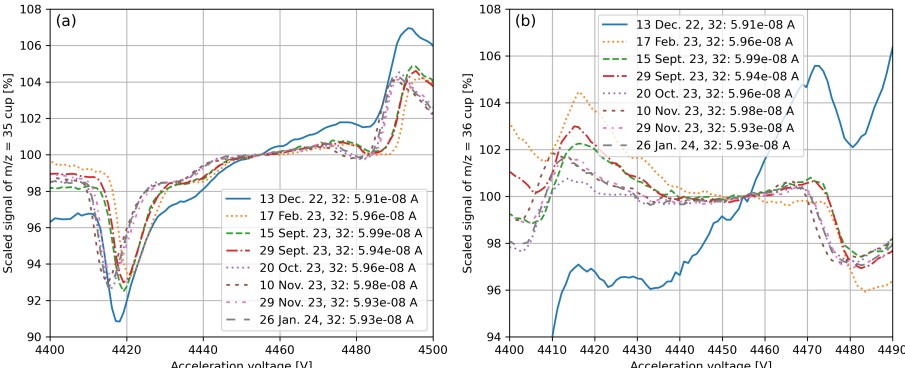

**Figure 8.** Scaled (a) $m/z = 35$ and (b) $m/z = 36$ peaks recorded during acceleration voltage scans of pure-oxygen gas (cylinder SC 540546) carried out on different days (see legend). The data are scaled versions of those presented in Fig. 7, with scaling based on the $m/z = 35$ and $m/z = 36$ signals at $U_{av} = 4455$ V, respectively.

background corrections; instead of strictly positive signals, meaningless negative values can be obtained. In such cases, not only is the ratio's value incorrect, but also its sign. Therefore, it is advisable to assess the background in the presence of the analyte, for instance, through the pressure baseline approach. In the following subsections, we showcase how pressure baseline
corrections can be determined, report different types as well as levels of correction and present the corresponding performance tests.

## 4.1 Correction procedure

To perform PBL corrections as suggested by Bernasconi (2013), AV scans at different signal intensities have to be performed to determine the relationships (correlations) between different on-peak and off-peak signals (background values). During con-
ventional SA-ST measurements, the SA and ST gas are measured multiple times in alternating order, but always at a single acceleration voltage (measurement position). Therefore, it is only possible to use signals recorded at the measurement position as predictors of the background; otherwise, additional measurements at different positions would be required (see the procedure by He (2012) outlined in the introduction).

In Table. 1, we present correlations between the signals left or right of clumped-isotope peaks and the signals recorded with
the $m/z = 30$, $m/z = 32$ and $m/z = 40$ cups (background predictors) at the measurement position. The positions left and right of the peaks were determined visually using different AV scans. It is also worth mentioning that we selected the $m/z = 30$ and $m/z = 40$ cups because they are located next to the cups used for measuring oxygen and thus might represent the background in this region most accurately. This selection is consistent with one of the methods suggested by Bernasconi (2013).

The coefficients of determination listed in Table 1 are all close to 1 (i.e., indicating high correlation). For both $m/z = 35$ and $m/z = 36$, the highest coefficients of determination were obtained for correlations with the $m/z = 35$ and $m/z = 36$ signals, respectively, evaluated at the measurement position (4455 V). This is particularly evident in the case of $m/z = 36$ (see rows 13



**Table 1.** Coefficients of determination for linear regressions calculated for correlations between different signals measured in pure-oxygen gas (cylinder SC 540546). These values were determined from three series of AV scans (pressure-decrease measurements) conducted on different days (October 20, 2023, November 10, 2023, and November 29, 2023). The gas was filled into the reference bellow of the iso DUAL INLET and then AV scans were periodically carried out while the pressure continuously decreased. All three measurement series covered $m/z = 32$ signals between $2 \cdot 10^{-8}$ A and $9 \cdot 10^{-8}$ A. The number of scans per measurement series were 69, 141 and 143, respectively. The abbreviations „BG left“ and „BG right“ refer to the positions where the background signals were determined, namely left and right of the peak, respectively. The positions used for this purpose were 4432 V for $m/z = 35$ BG left, 4481 V for $m/z = 35$ BG right, 4400 V or 4401 V for $m/z = 36$ BG left and 4480 V or 4481 V for $m/z = 36$ BG right. The other signals were evaluated at a common measurement position (4455 V).

| Signal on y-axis | Signal on x-axis | $R^2$ average [ ] | $R^2$ std. dev. [ ] |
|---|---|---|---|
| $m/z = 35$ BG left | $m/z = 30$ | 0.992 | $5 \cdot 10^{-3}$ |
| $m/z = 35$ BG right | $m/z = 30$ | 0.991 | $5 \cdot 10^{-3}$ |
| $m/z = 35$ BG left | $m/z = 32$ | 0.9983 | $9 \cdot 10^{-4}$ |
| $m/z = 35$ BG right | $m/z = 32$ | 0.9983 | $9 \cdot 10^{-4}$ |
| $m/z = 35$ BG left | $m/z = 35$ | 0.99977 | $2 \cdot 10^{-5}$ |
| $m/z = 35$ BG right | $m/z = 35$ | 0.99972 | $3 \cdot 10^{-5}$ |
| $m/z = 35$ BG left | $m/z = 40$ | 0.9984 | $9 \cdot 10^{-4}$ |
| $m/z = 35$ BG right | $m/z = 40$ | 0.998 | $1 \cdot 10^{-3}$ |
| $m/z = 36$ BG left | $m/z = 30$ | 0.97 | $4 \cdot 10^{-2}$ |
| $m/z = 36$ BG right | $m/z = 30$ | 0.97 | $4 \cdot 10^{-2}$ |
| $m/z = 36$ BG left | $m/z = 32$ | 0.97 | $4 \cdot 10^{-2}$ |
| $m/z = 36$ BG right | $m/z = 32$ | 0.97 | $5 \cdot 10^{-2}$ |
| $m/z = 36$ BG left | $m/z = 36$ | 0.9998 | $2 \cdot 10^{-4}$ |
| $m/z = 36$ BG right | $m/z = 36$ | 0.9999 | $2 \cdot 10^{-4}$ |
| $m/z = 36$ BG left | $m/z = 40$ | 0.97 | $5 \cdot 10^{-2}$ |
| $m/z = 36$ BG right | $m/z = 40$ | 0.97 | $5 \cdot 10^{-2}$ |

and 14 of Table 1). Although all coefficients of determination are close to the maximum, the performance of the corresponding corrections is appreciably different, as demonstrated in the next section (see Table 2). Through various tests documented in subsequent sections, we have established the following basic procedure for determining adequate pressure baseline corrections for peaks with linearly increasing or decreasing tops (example based on the $m/z = 35$ signal):

1. *Identification of adequate background positions.* From Fig. 9, which visualises our basic correction procedure, it can be seen that the $m/z = 35$ peak linearly increases with the acceleration voltage. Therefore, selecting positions (acceleration voltages) for determining the background on both the low- and high-mass side of the peak is required. For instance,



the AV scan depicted in Fig. 9 suggests that suitable positions for the determining the $m/z = 35$ background might be
4433 V and 4479 V; these positions may change over time, though (see Fig. 7). The variability of baselines left and right
of the peak has also been observed by other groups, e.g., Yeung (2018).

2. *Computation of correlations between the predictor and background signals.* First, AV scans have to be performed at
different signal intensities. We fill pure-oxygen gas into one of the iso DUAL INLET's bellows and periodically carry

out AV scans (usually every 30 min). Due to the steady consumption of gas, the signal intensity gradually decreases.
Typically, our measurement series consists of 50 to 100 AV scans and covers $m/z = 32$ signals ranging from $2 \cdot 10^{-8}$ A
to $9 \cdot 10^{-8}$ A. The SA-ST measurements to which the corrections are applied are normally conducted at $m/z = 32$ signal
intensities between $3 \cdot 10^{-8}$ A and $9 \cdot 10^{-8}$ A.

Next, the predictor signal (e.g., $m/z = 35$ evaluated at the measurement position of SA-ST measurements) and the

corresponding background signal left of the peak (e.g., $m/z = 35$ signal evaluated at 4436 V) have to be extracted from
each AV scan. Subsequently, correlations between the two signals can be computed. Finally, this procedure has to be
repeated for predicting the $m/z = 35$ signal's right background (e.g., correlation between $m/z = 35$ evaluated at the
measurement position of SA-ST measurements and $m/z = 35$ evaluated at 4484 V). To remove outliers, we discard
values that deviate from the original fit by more than 2 %. For highly correlated signals such as those mentioned in this

paragraph, this filter rarely removes any data points.

3. *Correction of individual measurement intervals.* Using the correlations determined in the previous step, the average
measurement signal recorded during an interval of a SA-ST measurement is used to estimate the background left and
right of the peak. We refer to these points as „BG left" and „BG right", respectively. These values are then interpolated
using a linear regression to estimate the background at the measurement position. From Fig. 10, it can be seen that

this PBL correction successfully reduces the influence of secondary electrons. While an increase in the (uncorrected)
$m/z = 32$ signal leads to a reduction of the uncorrected $m/z = 35$ signal, the PBL-corrected values both increase as a
function of the signal intensity.

It is worth mentioning that a linear interpolation of the background values is only appropriate if the peak tops are not curved;
all of our oxygen signals, except for $m/z = 36$, fulfil this condition. In Sect. 4.4.4, we discuss how to adapt our basic procedure

to correct $m/z = 36$ signals properly. Furthermore, it is noteworthy that panel (b) of Fig. 9 clearly shows that our correction
successfully generates a square-shaped $m/z = 35$ peak.

## 4.2   Comparison of different corrections

The experiment, whose results are presented in Table 2, was conducted to study how the 35/32 and $\delta_{35}$ values vary when the
$m/z = 35$ background is predicted using signals from different cups. These values were computed from a measurement series

consisting of 10 individual SA-ST measurements of pure-oxygen gas (12 SA and 13 ST intervals per measurement, 60 s idle
time and 20 s integration time). The pressure baseline corrections applied to the data were determined as outlined in Sect. 4.1.



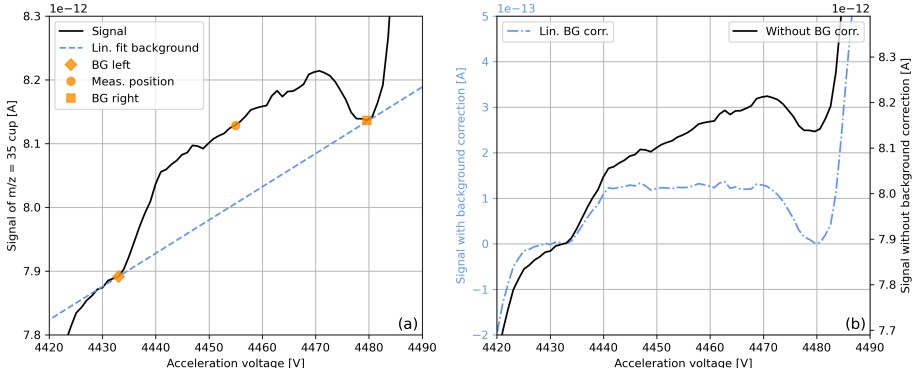

**Figure 9.** Uncorrected and pressure-baseline-corrected $m/z = 35$ signals recorded during an acceleration voltage scan of pure-oxygen gas (cylinder SC 540546). Panel (a) shows the raw $m/z = 35$ signal along with a linear regression line through two background points left and right of the peak (determined visually). Panel (b) contrasts the raw and pressure-baseline-corrected $m/z = 35$ signals. For comparability, the former signal was shifted to zero. The PBL correction follows the procedure described in Sect. 4.1, where the linear pressure baseline shown in (a) is subtracted from the raw signals. In addition, panel (a) indicates a common measurement position for SA-ST measurements, which is used to predict the signal at the two background positions.

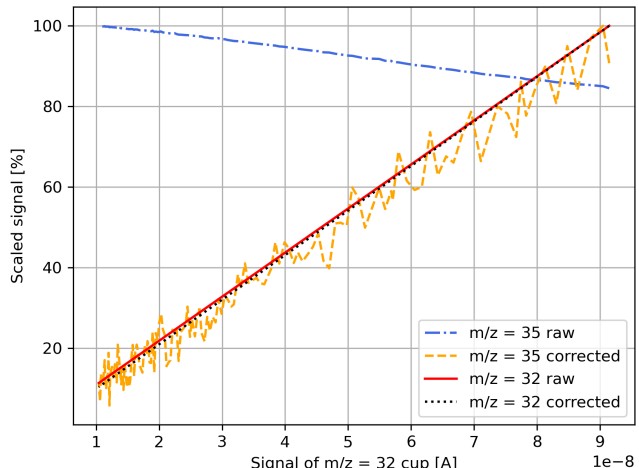

**Figure 10.** Uncorrected and pressure-baseline-corrected $m/z = 32$ and $m/z = 35$ signals measured at 4455 V. The values were extracted from a series of acceleration voltage scans of pure-oxygen gas (cylinder SC 540546) performed at different signal intensities ($m/z = 32$ signals between $2 \cdot 10^{-8}$ A and $9 \cdot 10^{-8}$ A). The gas was filled into the reference bellow of the iso DUAL INLET and then a measurement was performed every 30 min. Due to the steady consumption of gas, the signal gradually decreased. The PBL corrections were determined as described in Sect. 4.1 (see enumeration). All signals were scaled with respect to their maximum.

Most importantly, these data indicate that the uncertainties of the isotope ratios are smallest when the $m/z = 35$ signal's background is predicted using the $m/z = 35$ peak itself. The second-best correction is provided by the $m/z = 40$ signal, which



**Table 2.** Averages of pressure-baseline-corrected 35/32 and $\delta_{35}$ values recorded during 10 consecutive SA-ST measurements ($U_{av} = 4455$ V) of pure-oxygen gas (SA cylinder SC 540546 and ST cylinder SC 62349). The gas was admitted to the Elementar isoprime precisION through the NIS-II. An electron suppressor voltage of -140 V was applied and the $m/z = 32$ signal was around $8.3 \cdot 10^{-8}$ A. Per measurement, 13 ST and 12 SA intervals were recorded. The integration time was 20 s. The indicated uncertainties correspond to the standard deviations of all independent measurement intervals recorded during the entire measurement series (130 values for 35/32 ST and 120 values for 35/32 SA, as well as $\delta_{35}$). The PBL corrections were determined as described in Sect. 4.1. The predictor for the left and right background of the $m/z = 32$ signal was its on-peak signal. The predictors for the two background values of the $m/z = 35$ signal are indicated in the first column of the table. All predictor signals were evaluated at $U_{av} = 4455$ V.

| Predictor of $m/z = 35$ BG [A] | 35/32 ST [ ] | 35/32 SA [ ] | $\delta_{35}$ [‰] |
|---|---|---|---|
| $m/z = 30$ | $1.144 \cdot 10^{-4} \pm 1 \cdot 10^{-7}$ | $1.148 \cdot 10^{-4} \pm 1 \cdot 10^{-7}$ | $4.0 \pm 0.5$ ‰ |
| $m/z = 32$ | $1.01203 \pm 2 \cdot 10^{-5}$ | $1.01193 \pm 2 \cdot 10^{-5}$ | $-0.10 \pm 0.01$ ‰ |
| $m/z = 35$ | $1.33 \cdot 10^{-6} \pm 1 \cdot 10^{-8}$ | $1.33 \cdot 10^{-6} \pm 1 \cdot 10^{-8}$ | $-0.3 \pm 0.2$ ‰ |
| $m/z = 40$ | $-1.517 \cdot 10^{-5} \pm 2 \cdot 10^{-8}$ | $-1.512 \cdot 10^{-5} \pm 2 \cdot 10^{-8}$ | $-3.4 \pm 0.2$ ‰ |

yields an uncertainty that is approximately twice as high. At first glance, the correction based on the $m/z = 32$ peak appears

to work best for $\delta_{35}$. However, as can be deduced from the equations in the appendix of Räss (2023), the indicated precision is misleading because the corresponding isotope ratios are much larger than the others, resulting in smaller uncertainties.

If the collector zero correction is applied to the $m/z = 32$ signal instead of the PBL correction, the 35/32 ST, 35/32 SA and $\delta_{35}$ values are indistinguishable from those displayed in Table 2 within the measurement uncertainties. When the $m/z = 35$ signal is also corrected using the collector zero value, the averages for 35/32 ST, 35/32 SA and $\delta_{35}$ are $-2.04 \cdot 10^{-5} \pm 2 \cdot 10^{-7}$,

$-2.04 \cdot 10^{-5} \pm 2 \cdot 10^{-7}$ and $0.1 \pm 0.2$ ‰, respectively. These values cannot be correct, as isotope ratios should always be strictly positive.

To test the robustness of our evaluation, we also calculated the uncertainties indicated in Table 2 for 30 % and 60 % of the entire data set. The standard deviations of the uncertainties of the 35/32 ratios for the two subsets and the full data set are all approximately 1 order of magnitude lower than the uncertainties indicated in Table 2; for $\delta_{35}$, these standard deviations are

about 2 orders of magnitude lower.

The values listed in Table 3 show how 35/32 and $\delta_{35}$ vary when the same PBL correction is applied to all measurement intervals. For one of these corrections, we used the average of the $m/z = 35$ signals from the entire measurement series to calculate a single background value. For the second PBL correction, we computed separate values for the SA and ST intervals by averaging all SA and ST intervals, respectively. Apart from these differences, the determination of the background follows

the principles described in Sect. 4.1.

Although all of the corrections lead to positive isotope ratios, the values shown in Table 3 clearly indicate that considering the variability of measurement intervals makes a difference. For the data set at hand, the uncertainties of the isotope ratios and delta values were reduced by more than 1 order of magnitude. Moreover, using the left background value leads to an





**Table 3.** Averages of pressure-baseline-corrected 35/32 and $\delta_{35}$ values recorded during 10 consecutive SA-ST measurements ($U_{av} = 4455$ V) of pure-oxygen gas (SA cylinder SC 540546 and ST cylinder SC 62349). The measurements and calculations of uncertainty were performed as described in the caption of Table 2. The ratios were corrected using different pressure baseline corrections. The $m/z = 32$ signal was corrected as described in Sect. 4.1. The method used for the $m/z = 35$ signal correction is indicated in the first column. The term „left BG" („right BG") denotes that the correlation between the $m/z = 35$ signal and its left (right) background was used. The term „signal average" refers to using the average of all $m/z = 35$ signals as the predictor and „SA/ST" indicates that a distinction was made between SA and ST intervals (i.e., two predictors instead of one). In the last row, the same data are shown as in the third row of Table 2. The data in the second last row were corrected as the data in the last row with the difference that only the left BG value was subtracted from each measurement interval (no interpolation).

| Correction | 35/32 ST [ ] | 35/32 SA [ ] | $\delta_{35}$ [‰] |
|---|---|---|---|
| signal average, left BG | $2.4 \cdot 10^{-6} \pm 2 \cdot 10^{-7}$ | $2.2 \cdot 10^{-6} \pm 2 \cdot 10^{-7}$ | $-84 \pm 14\,‰$ |
| signal average, right BG | $3 \cdot 10^{-7} \pm 2 \cdot 10^{-7}$ | $1 \cdot 10^{-7} \pm 2 \cdot 10^{-7}$ | $-1415 \pm 5687\,‰$ |
| SA/ST signal average, left BG | $2.3 \cdot 10^{-6} \pm 2 \cdot 10^{-7}$ | $2.3 \cdot 10^{-6} \pm 2 \cdot 10^{-7}$ | $0 \pm 12\,‰$ |
| SA/ST signal average, right BG | $2 \cdot 10^{-7} \pm 2 \cdot 10^{-7}$ | $2 \cdot 10^{-7} \pm 2 \cdot 10^{-7}$ | $-257 \pm 5595\,‰$ |
| individual intervals, left BG | $2.30 \cdot 10^{-6} \pm 2 \cdot 10^{-8}$ | $2.30 \cdot 10^{-6} \pm 2 \cdot 10^{-8}$ | $-0.1 \pm 0.2\,‰$ |
| individual intervals | $1.33 \cdot 10^{-6} \pm 1 \cdot 10^{-8}$ | $1.33 \cdot 10^{-6} \pm 1 \cdot 10^{-8}$ | $-0.3 \pm 0.2\,‰$ |

underestimation of the actual background value, while the right background leads to an overestimation. The reason is that the
$m/z = 35$ signal increases as a function of the acceleration voltage. Furthermore, note that applying collector zero corrections results in the same problems, as a single value is subtracted from the signals. Additionally, comparing the last two rows of Table 3 shows that correcting the data by subtracting the minimum background value computed for each interval individually still results in significantly higher uncertainty compared to the full correction. In the case at hand, the minimum value was the background left of the peak. This example further highlights that the procedure suggested by Bernasconi (2013) needed to be
modified to improve our results. Unless otherwise stated, all subsequent PBL corrections are applied to individual measurement intervals and consider both left and right BG values.

In addition to the corrections reported in Table 3, we also calculated 35/32 and $\delta_{35}$ averages for the case where only the $m/z = 35$ signal is PBL-corrected and the $m/z = 32$ signal is corrected using the collector zero value. No significant differences were observed within the measurement uncertainties. From this, we conclude that for our data, $m/z = 32$ signals can be
corrected using the collector zero value and that the uncertainty is dominated by $m/z = 35$.

## 4.3 Drift correction

Although PBL corrections account for signal variations, signal drifts may still remain. Therefore, we tested whether applying additional drift corrections could improve our results. For this correction, we separated the ST intervals of the 10 measurements



**Table 4.** Averages of pressure-baseline-corrected 35/32 and $\delta_{35}$ values recorded during 10 consecutive SA-ST measurements ($U_{av} = 4455$ V) of pure-oxygen gas (SA cylinder SC 540546 and ST cylinder SC 62349). The measurements and calculations of uncertainty were carried out as outlined in the caption of Table 2. The PBL correction was applied to the individual intervals of the numerator and denominator components of the isotope ratio (see Sect. 4.1). The data reported in the second and third rows of the table were corrected using linear and polynomial drift corrections, respectively. These corrections were applied separately to the PBL-corrected SA and ST ratios.

| Drift correction | 35/32 ST [ ] | 35/32 SA [ ] | $\delta_{35}$ [‰] |
|---|---|---|---|
| none | $1.33 \cdot 10^{-6} \pm 1 \cdot 10^{-8}$ | $1.33 \cdot 10^{-6} \pm 1 \cdot 10^{-8}$ | $-0.3 \pm 0.2$ ‰ |
| linear | $1.308 \cdot 10^{-6} \pm 1 \cdot 10^{-9}$ | $1.307 \cdot 10^{-6} \pm 1 \cdot 10^{-9}$ | $-0.3 \pm 0.2$ ‰ |
| polynomial | $1.3057 \cdot 10^{-6} \pm 8 \cdot 10^{-10}$ | $1.3053 \cdot 10^{-6} \pm 8 \cdot 10^{-10}$ | $-0.4 \pm 0.2$ ‰ |

from the SA intervals and regressed these intervals on the interval index separately. We defined the first interval as the point of reference and corrected the subsequent intervals according to the corresponding regressions. The results listed in Table 4 point out that applying such drift corrections can markedly reduce the variability of the isotope ratios. In our data, the linear correction reduced the uncertainty by approximately 1 order of magnitude, while the polynomial correction led to an additional improvement of roughly 20 %. Applying the drift corrections directly to the signals instead of the ratios yielded similar results. In contrast to the uncertainty of the isotope ratios, the drift correction did not significantly reduce the uncertainty of the delta values (see Table 4). Nevertheless, this is reasonable because the drifts of SA and ST ratios are similar and ultimately cancel out.

Comparing the uncertainties of the drift-corrected 35/32 and $\delta_{35}$ values to those reported by Laskar (2019) (their supplement) shows that our standard deviations are all lower when a polynomial drift correction is applied – even though we evaluated twice as many values. Laskar (2019) used a Thermo Scientific 253 Ultra High Resolution IRMS to measure clumped isotopes of oxygen extracted from atmospheric air and obtained standard deviations around $1 \cdot 10^{-9}$ and $0.9$ ‰ for 35/32 and $\delta_{35}$, respectively. In contrast, the uncertainties we achieved with the polynomial drift correction are approximately $8 \cdot 10^{-10}$ and $0.2$ ‰, respectively (see Table 4). Applying a linear drift correction instead, the standard deviations of 35/32 are similar to those reported by Laskar (2019).

## 4.4 Correction of capital delta values

Typically, the most prominent quantity in clumped-isotope studies is the capital delta value defined in Eq. (1). In this subsection, we address different aspects of $\Delta_{35}$ and $\Delta_{36}$ related to PBL corrections.

### 4.4.1 Evaluation of $\Delta_{35}$

When calculating $\Delta_{35}$ from the data discussed in Sect. 4.2 (10 individual SA-ST measurements of pure-oxygen gas) and predicting the $m/z = 32$ as well as $m/z = 35$ backgrounds using the corresponding on-peak intensities, an average of $-2.0 \pm$





0.5 ‰ is obtained. The uncertainty corresponds to $1\sigma$ and was calculated from 120 values, which is nearly half the standard deviation reported by Laskar (2019) (their supplement) for 60 values. Within the measurement uncertainty, the results obtained from collector-zero-corrected and PBL-corrected $m/z = 32$ signals are indistinguishable. Moreover, we repeated the analysis using a collector zero value determined several months later and came to the same conclusion.

For the calculation of $\Delta_{35}$ (see Eq. (1)), the stochastic ratio $R^*$ ($^{35}R^*_{SA}$) was estimated using the averages of the corresponding collector-zero-corrected 34/32 and 33/32 ratios. Furthermore, $R$ ($^{35}R_{SA}$) was deduced from the measured $\delta_{35}$ average and a fixed value for the ST, which was used for all measurements of the series. The corresponding formula can be derived by rearranging the terms of Eq. (2) as follows:

$$^{35}R_{SA} \;=\; {}^{35}R_{ST} \cdot \left( \frac{\delta_{35}(‰)}{1000} + 1 \right). \tag{3}$$

Fixing the value of the ST ratio is a common procedure (Huntington, 2023) (their supplement), which may help to account for the instrument's drift. For this fixed ratio, we used the 35/32 ratio derived from the standard intervals of 33/32 and 34/32 (see Appendix A).

   If the ratio is not fixed, the standard deviation of the isotope ratios and capital delta values increases from $3 \cdot 10^{-10}$ to $1 \cdot 10^{-8}$ and from 0.5 ‰ to 8 ‰, respectively. When a drift correction is applied to the data before calculating the individual $\Delta_{35}$ values,

the average is $-2.0 \pm 0.4$ ‰ ($1\sigma$ uncertainty of 120 values). Furthermore, if a single $R^*$ value is used for all individual values, the uncertainty reduces to 0.2 ‰.

### 4.4.2   Non-linearity

As mentioned in the introduction, various research groups have observed an apparent dependence of $\Delta_{47}$ on $\delta_{47}$ (non-linearity), which can be reduced through PBL corrections (Bernasconi, 2013). When calculating $\Delta_{35}$ from our collector-zero-corrected

data (stochastic SA ratio calculated for each interval individually), plotting those values against $\delta_{35}$ and modelling the relationship using a linear fit, the coefficient of determination is around 0.29. We repeated the calculation for PBL-corrected data and obtained a coefficient of determination around 0.73. Hence, applying a PBL correction to our data led to an even higher correlation. Since this was unexpected, we investigated this issue and concluded that the non-linearity may actually be due to the relatively large discrepancy between the uncertainties of $R$ and $R^*$. The data presented in Sect. 4.2 indicate that the standard

deviation of the PBL-corrected $^{35}R$ ratios is of the order $1.2 \cdot 10^{-9}$ if a drift correction is applied and $1.17 \cdot 10^{-8}$ otherwise. In contrast, the standard deviation of $^{35}R^*$ is merely on the order of $6 \cdot 10^{-10}$. Consequently, $^{35}R$ divided by $^{35}R^*$ (and thus $\Delta_{35}$) varies as a function of $^{35}R$, which results in a strong correlation between the delta and capital delta values (see Fig. 11).

### 4.4.3   Uncertainty of $\Delta_{35}$

Since our PBL corrections are computed from various components, which are in turn associated with different uncertainties, we analysed their influence on the uncertainty of $\Delta_{35}$ through Monte Carlo simulations.




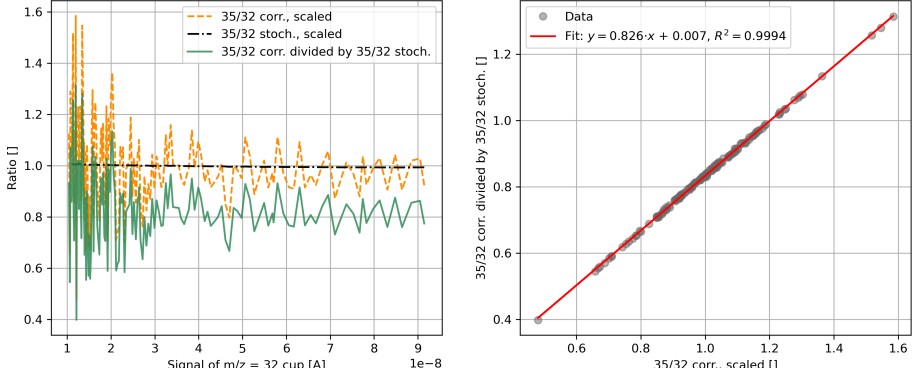

**Figure 11.** (a) Stochastic (stoch.) and pressure-baseline-corrected (corr.) 35/32 ratios measured in pure-oxygen gas (cylinder SC 540546), which were scaled with respect to the corresponding averages. Additionally, the former ratio divided by the latter is depicted. The presented values were calculated from a series of acceleration voltage scans performed at different signal intensities ($m/z = 32$ signals between $2 \cdot 10^{-8}$ A and $9 \cdot 10^{-8}$ A) and evaluated at $U_{av} = 4465$ V. The measurement procedure is detailed in the caption of Fig. 10. The pressure baseline corrections and the stochastic ratios were determined as described in Sect. 4.1 and Sect. 4.4.1, respectively. Panel (b) shows the correlation of the corrected 35/32 ratios and the corrected 35/32 ratios divided by the stochastic 35/32 ratios (same data as in panel (a)); it also includes the corresponding linear regression and its coefficient of determination.

First, we determined the quantities required for the calculation of corrected $\Delta_{35}$ values (see Eq. (1), Eq. B2 and Eq. B6). Next, we varied one of these quantities at a time while keeping the others constant. For each quantity, we generated $10^6$ samples from a normal distribution. The centre of the distribution was set to the average recorded during our SA-ST measurement
series (see Sect. 4.2) and its standard deviation was set to 1 permil of that value. This simulation yielded the $\Delta_{35}$ uncertainties listed in Table 5. From these values, we conclude that the contributions from the $m/z = 32$ background values and the drift corrections are negligible. The main contribution is associated with the raw $m/z = 35$ on-peak signals and its background values. The contributions from the 33/32 and 34/32 ratios (required for calculating stochastic 35/32 ratios), as well as those from the raw $m/z = 32$ on-peak signals, are small but not insignificant.
Varying the raw $m/z = 35$ on-peak signal using its measured standard deviation as the standard deviation of the normal distribution, the simulations indicate that the relative deviation of the simulated $\Delta_{35}$ value from the measured value is around $-4 \cdot 10^{-3}$ %. The standard deviation of the simulated values relative to the measured average is roughly 18 %. Varying the left and right $m/z = 35$ backgrounds resulted in relative deviations of the simulated $\Delta_{35}$ values from the measured value of approximately $-1 \cdot 10^{-2}$ % and $5 \cdot 10^{-3}$ %, respectively. The corresponding standard deviations of the simulated values relative
to the measured averages were about 10 % and 8 %, respectively. For the simulations, we used $3.4 \cdot 10^{-15}$ A (raw $m/z = 35$ on-peak signal), $3.8 \cdot 10^{-15}$ A (left $m/z = 35$ background) and $3.4 \cdot 10^{-15}$ A (right $m/z = 35$ background) as uncertainties, which correspond to the standard deviations of 120 drift-corrected values. For the other contributors, the relative deviations of the simulated $\Delta_{35}$ values from the measured value were all smaller than $4 \cdot 10^{-3}$ %.





**Table 5.** Relative deviations of simulated $\Delta_{35}$ values from the measured average, as well as standard deviations of simulated $\Delta_{35}$ values relative to the measured average. The $\Delta_{35}$ values were obtained through Monte Carlo simulations. In these simulations, one of the parameters used in the calculation of $\Delta_{35}$ was varied (as indicated in the first column), while all other parameters were fixed. For each simulation, $1 \cdot 10^6$ samples were drawn from a normal distribution. The centre of the distribution corresponded to the average of the varied parameter, while the standard deviation was 1 permil of that value.

| Varied parameter | Relative deviation from average [%] | Relative standard deviation [%] |
|---|---|---|
| raw $m/z = 35$ on-peak signal | $-2 \cdot 10^{-2}$ | 43 |
| left $m/z = 35$ background | $2 \cdot 10^{-2}$ | 22 |
| right $m/z = 35$ background | $-2 \cdot 10^{-2}$ | 20 |
| drift corr. of $m/z = 35$ | $-3 \cdot 10^{-6}$ | $9 \cdot 10^{-3}$ |
| raw $m/z = 32$ on-peak signal | $1 \cdot 10^{-3}$ | $6 \cdot 10^{-1}$ |
| left $m/z = 32$ background | $4 \cdot 10^{-6}$ | $6 \cdot 10^{-3}$ |
| right $m/z = 32$ background | $6 \cdot 10^{-7}$ | $1 \cdot 10^{-3}$ |
| drift corr. of $m/z = 32$ | $-1 \cdot 10^{-8}$ | $2 \cdot 10^{-4}$ |
| 33/32 | $8 \cdot 10^{-4}$ | $6 \cdot 10^{-1}$ |
| 34/32 | $1 \cdot 10^{-4}$ | $6 \cdot 10^{-1}$ |

Our simulations clearly demonstrate that minor variations in the previously mentioned contributions can result in significant fluctuations in the $\Delta_{35}$ average. In addition to the Monte Carlo simulations, we also analysed the contributions using the propagation of uncertainty (see Appendix B) and came to the same conclusions. Unlike the Monte Carlo simulation, the latter approach also allows for the analysis of the influence of correlation terms, which is non-negligible due to high correlations between certain terms (e.g., between the on-peak signal and predicted backgrounds).

It is worth noting that we did neglect the uncertainties associated with the BG positions and the regressions used for the prediction of the BG values. Not only does the background change over time (see Fig. 8), but it also depends on the signal intensity (Fig. 7). Moreover, the visual determination of background positions is somewhat subjective and is associated with an uncertainty of roughly $\pm 1$ V. Furthermore, $\Delta_{35}$ depends on the value of the fixed 35/32 standard ratio. Hence, considering these factors the uncertainty for the measured $\Delta_{35}$ values might actually be higher than 0.5 permil.


### 4.4.4    Adjustment of PBL correction for $m/z = 36$ signals

While our procedure for correcting $m/z = 35$ signals is already more involved than the PBL corrections presented by He (2012) and Bernasconi (2013), $m/z = 36$ peaks introduce an additional layer of complexity due to their curvature (see Fig. 8). Specifically, from Fig. 12, it can be seen that the curvature of the $m/z = 36$ peaks changes with signal intensity and that the
linear correction overestimates the background.





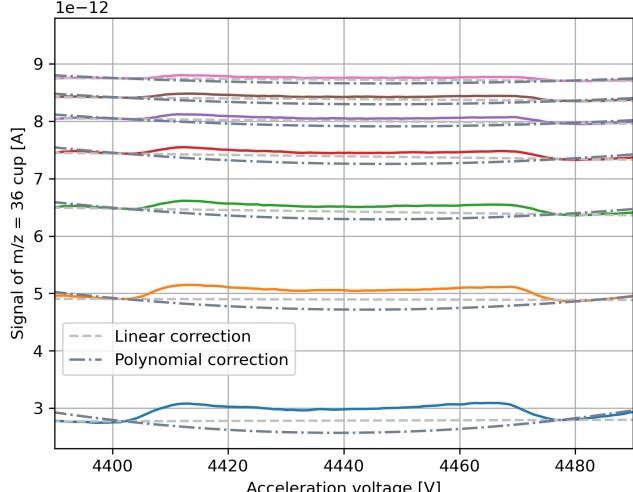

**Figure 12.** Uncorrected signals recorded during a series of acceleration voltage scans of pure-oxygen gas (cylinder SC 540546) on the $m/z = 36$ cup. The scans were performed at $m/z = 32$ signal intensities between $2 \cdot 10^{-8}$ A and $9 \cdot 10^{-8}$ A. The gas was filled into the reference bellow of the iso DUAL INLET and then measurements were performed every 30 min. Due to the steady consumption of gas, the signal gradually decreased. In addition to the uncorrected data, linear and second-order polynomial fits are displayed. Essentially, these fits were calculated following the procedure described in Sect. 4.4.4, with the distinction that acceleration voltage scans were processed instead of SA-ST measurements.

From our AV scans it can be seen that the curvature of the $m/z = 36$ signal's peak top might actually be induced by the background, as it appears to be negatively curved as well. Additionally, this aligns with the statements made by He (2012). To account for the aforementioned curvature, in Fig. 12, we show second-order polynomials that may represent the background more appropriately than the linear background fits depicted in the same figure. In contrast to our basic approach presented in

Sect. 4.1, for this type of PBL correction, at least three correlations are required instead of two. In addition to the correlation between the measured $m/z = 36$ on-peak signal and the left (or right) background position, correlations between the measured $m/z = 36$ on-peak signal and two other points on the peak top have to be determined. Typically, the predictor signal is measured in the central region of the peak top, whereas the predicted signals are usually situated close to the left and right edges of the peak top, respectively; thus, we refer to these points as „PT left" and „PT right". As a next step, the measured on-peak signal

and the two predicted signals are used to compute a second-order polynomial fit modelling the peak top. Subsequently, it has to be determined by how much the peak top fit should be shifted downwards to become an appropriate background model. For instance, one option is to determine the difference between the linear fit going through the two BG points and the peak top fit at the position of the left BG (see „shift (1)" in Fig. 13). Alternatively, the peak top fit could be shifted by the difference between the left PT and the left BG point (see „shift (2)" in Fig. 13). In Fig. 13, we also indicate shifts denoted as „shift (3)" and „shift

(4)", which are the equivalents of the aforementioned positions but for the right side of the peak.





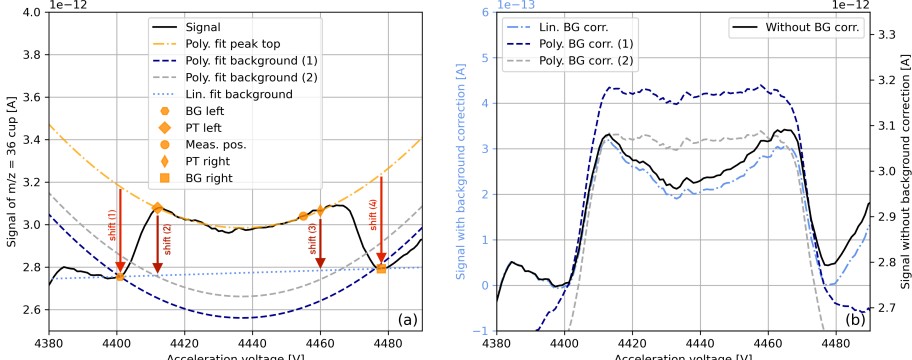

**Figure 13.** Uncorrected and pressure-baseline-corrected $m/z = 36$ signals recorded during an acceleration voltage scan of pure-oxygen gas (cylinder SC 540546). In panel (a), the raw $m/z = 36$ signal is shown along with a linear regression going through two background points left and right of the peak (determined visually). In the same panel, a second-order polynomial regression is shown, obtained by fitting the three points indicated on the peak top. This regression was then shifted downwards by (1) the difference between the polynomial regression going through the peak top (PT) and the linear fit evaluated at the left background (BG) point, as well as by (2) the difference between the signals evaluated at the left PT and left BG position. The shifts indicated as (3) and (4) are the equivalents of (2) and (1), respectively, but for the right side of the peak. In panel (b), raw and pressure-baseline-corrected $m/z = 36$ signals are contrasted. For one of the PBL corrections, the linear regression (lin. fit background) depicted in panel (a) was subtracted from the raw data and for the others, the polynomial regressions (poly. fit background (1) and (2)). Essentially, the determination of these corrections follows the procedures described in Sect. 4.1 and Sect. 4.4.4, but using AV scans rather than SA-ST measurements.

As can be seen from panel (a) of Fig. 12, the former method generates a pressure baseline going through the bottom of the peak. However, panel (b) of this figure also shows that the polynomially-corrected peak appears to be higher than the original peak, indicating that the actual background might be slightly less curved than the peak top. In contrast, the method applying „shift (2)“ seems to generate a peak of proper height, but the corresponding pressure baseline appears to be too high (see Fig. 13). Yet, we have not examined this issue in detail. The reason is that primarily accuracy rather than precision is affected (see Sect. 4.4.5). Although the polynomial PBL correction is more difficult to control than the linear PBL correction, Fig. 12 highlights that the linear correction is not adequate because it cannot eliminate the curvature, resulting in a clear dependence of the signal intensity on the measurement position.

Apart from the determination of the polynomial background fit, pressure baseline corrections are applied to individual measurement intervals of SA-ST measurements as described in Sect. 4.1.

### 4.4.5 Evaluation of 36/32, $\delta_{36}$ and $\Delta_{36}$

Based on the procedures outlined in the previous section, we applied different PBL corrections to both $m/z = 32$ and $m/z = 36$ signals recorded during the SA-ST-measurement series, which consisted of 10 individual measurements of pure-oxygen gas (see Sect. 4.2). The results shown in Table 6 indicate that for 36/32 we obtained precisions as high as $1 \cdot 10^{-9}$, which



translates into an uncertainty of $\delta_{36}$ and $\Delta_{36}$ around $0.1\,\text{‰}$. To obtain these values, we applied a polynomial drift and pressure baseline correction to the raw data. For the latter correction, the curved peak top was modelled using a polynomial fit, which was shifted downwards by the difference between the left peak top point and the left background point; except for the on-peak signals, all points involved in this correction were predicted using signal correlations. Furthermore, Table 6 highlights that the uncertainties of the delta and capital delta values are quite consistent, ranging between $0.1\,\text{‰}$ and $0.2\,\text{‰}$ (230 values computed

from 10 SA-ST measurements with 12 SA and 13 ST intervals). Nonetheless, Table 6 also indicates that the type of correction can significantly influence the average and uncertainty of the isotope ratio; the largest discrepancy we observed is around 1 order of magnitude. Furthermore, it is noteworthy that the 36/32 average obtained with the linear correction is roughly 1 order of magnitude above the natural abundance (Meija, 2016; Räss, 2023). The variations in the 36/32 ratios resulted in $\delta_{36}$ averages in the range of -0.6 permil to 1.2 permil. Excluding the second row of Table 6, the $\Delta_{36}$ averages were relatively consistent, ranging between -2.7 permil and -2.2 permil. However, without proper calibration, external reproducibility cannot be assessed

effectively. Moreover, our data indicate that the selection of the point at which the downward shift of the polynomial peak top fit is determined can considerably influence the absolute value of $\Delta_{36}$ (compare the second row of Table 6 to other rows). Therefore, calibrating our values by measuring gases with known isotopic composition or gases conforming to the stochastic distribution (i.e., with a capital delta value equal to zero) is required.

As discussed in Sect. 4.4.4, the general procedure for determining PBL corrections for $m/z = 36$ peaks involves computing a polynomial fit from the measurement signal and two predicted points on the peak top. Since the the peak top of the $m/z = 36$ signal usually fluctuates considerably across its width, small variations in the two predicted points could substantially influence the fit. To assess whether a larger number of peak top points leads to more stable results, we predicted additional points between PT left and PT right of the $m/z = 36$ peaks, using the $m/z = 36$ on-peak signals recorded during our 10 SA-ST

measurements as predictors. These predictions were made in steps of 1 V (AV) and then the polynomial fits and shifts were computed as usual. The results shown in the last row of Table 6 imply that there was no significant difference in $\delta_{36}$ and $\Delta_{36}$ when compared to the method using three points for the calculation of the fit. Although the two approaches led to slightly different results for 36/32, the variation is well within the uncertainty associated with the downward shift of the fit. Due to these results and for the sake of simplicity, we decided to continue using the three-point method for the analyses presented in

this work. Please also note that for the polynomial PBL correction of the $m/z = 32$ signals, we never considered using more than three points, as variations of peak top signals relative to the height of the peak are much smaller and because it allows for studying the influence of $m/z = 36$ alone.

Comparing our results to those reported by Laskar (2019) (their supplement) shows that they obtained higher standard

deviations for $\delta_{36}$ and $\Delta_{36}$ than we did, namely around $0.7\,\text{‰}$ (60 values) for both of these values. For 36/32, they achieved a standard deviation around $4 \cdot 10^{-9}$, which is slightly inferior to our best results.

Although our study indicates that polynomial PBL corrections, along with polynomial drift corrections, might yield the most accurate and precise results, for subsequent studies, we generally skip the drift correction and use the linear PBL correction





**Table 6.** Averages of 36/32 SA, $\delta_{36}$ and $\Delta_{36}$ recorded during 10 consecutive SA-ST measurements ($U_{av} = 4455$ V) of pure-oxygen gas (SA cylinder SC 540546 and ST cylinder SC 62349). The measurements were carried out as outlined in the caption of Table 2. The indicated uncertainties correspond to the standard deviations of all intervals recorded during the entire measurement series (120 values for 36/32 and 230 values for $\delta_{36}$, as well as $\Delta_{36}$). The raw data were corrected as indicated in the first four columns. Pressure baseline corrections were applied to both $m/z = 36$ and $m/z = 32$ signals, following the explanations given in Sect. 4.4.4 and Sect. 4.1. Drift corrections were applied to 36/32 SA and 36/32 ST individually. The column denoted as „shift" indicates how linear (lin.) or polynomial (ply.) regressions were shifted to obtain suitable pressure baseline corrections: For „lin. minus ply. at meas. pos.", a single AV scan was used to estimate the difference between the linear and polynomial regression (both going through the background points) at the measurement position (same estimate subtracted from all measurement intervals). For „top ply. minus lin. at left BG", the difference between the peak top fit and the linear fit was evaluated at the left background (BG) position (see „shift (1)" in Fig. 13). Additionally, the difference between the left peak top (PT) point and the left background point was used as a shift, which is denoted as „left PT minus left BG" (see „shift (2)" in Fig. 13). For the latter two corrections, the differences were assessed for each interval individually. The same BG and PT positions were used for the entire measurement series. The capital delta values were calculated as described in Sect. 4.4.1, using the averages of the stochastic 36/32 SA and 36/32 ST ratios as fixed values, which were determined for each measurement individually.

| PBL correction | Fit points | Shift | Drift corr. | 36/32 SA [ ] | $\delta_{36}$ [%] | $\Delta_{36}$ [%] |
|---|---|---|---|---|---|---|
| linear | 2 | none | none | $2.541 \cdot 10^{-5} \pm 5 \cdot 10^{-9}$ | $-0.2 \pm 0.1$ | $-2.3 \pm 0.1$ |
| polynomial | 3 | top ply. minus lin. at right BG | none | $6.73 \cdot 10^{-6} \pm 1 \cdot 10^{-8}$ | $1.2 \pm 0.2$ | $-4.4 \pm 0.2$ |
| polynomial | 3 | top ply. minus lin. at left BG | none | $4.84 \cdot 10^{-6} \pm 1 \cdot 10^{-8}$ | $-0.4 \pm 0.1$ | $-2.6 \pm 0.1$ |
| polynomial | 3 | right PT minus right BG | none | $3.333 \cdot 10^{-6} \pm 7 \cdot 10^{-9}$ | $-0.6 \pm 0.1$ | $-2.7 \pm 0.1$ |
| polynomial | 3 | left PT minus left BG | none | $3.184 \cdot 10^{-6} \pm 7 \cdot 10^{-9}$ | $0.0 \pm 0.1$ | $-2.2 \pm 0.1$ |
| polynomial | 3 | left PT minus left BG | linear | $3.172 \cdot 10^{-6} \pm 2 \cdot 10^{-9}$ | $0.0 \pm 0.1$ | $-2.2 \pm 0.1$ |
| polynomial | 3 | left PT minus left BG | polynomial | $3.170 \cdot 10^{-6} \pm 1 \cdot 10^{-9}$ | $-0.1 \pm 0.1$ | $-2.2 \pm 0.1$ |
| polynomial | 53 | left PT minus left BG | polynomial | $3.234 \cdot 10^{-6} \pm 1 \cdot 10^{-9}$ | $-0.0 \pm 0.1$ | $-2.2 \pm 0.1$ |

instead. There are two reasons for this: first, the linear correction generally provides good estimates of $\delta_{36}$, as well as $\Delta_{36}$

values and second, the variations induced by factors other than curvature are easier to assess.

### 4.4.6 Uncertainty of $\Delta_{36}$

In analogy with the procedure presented in Sect. 4.4.3, we conducted Monte Carlo simulations to study how the total uncertainty of $\Delta_{36}$ is affected by the components involved in its calculation. For $m/z = 32$, the PBL correction was performed as described in Sect. 4.1. To assess the influence of the two background points and the curvature of the $m/z = 36$ signal's peak top on the

uncertainty of $\Delta_{36}$ separately, we did not apply a polynomial PBL correction to the $m/z = 36$ signal; instead, we adjusted the linear correction by adding a fixed value that accounts for the curvature. Hereafter, we refer to this correction as „fixed curvature correction" because it does not take intensity fluctuations into account. Please note that in Eq. (B2), which shows



**Table 7.** Relative deviations of simulated $\Delta_{36}$ values from the measured average, as well as standard deviations of simulated $\Delta_{36}$ values relative to this average. The values were generated through Monte Carlo simulations performed in analogy with Table 5 (see Sect. 4.4.3). The computation of $\Delta_{36}$ follows the explanations given in Sect. 4.4.6.

| Varied parameter | Relative deviation from average [%] | Relative standard deviation [%] |
|---|---|---|
| raw $m/z = 36$ on-peak signal | $-6 \cdot 10^{-3}$ | 19 |
| left $m/z = 36$ background | $1 \cdot 10^{-4}$ | 6 |
| right $m/z = 36$ background | $-2 \cdot 10^{-2}$ | 12 |
| curvature corr. $m/z = 36$ | $1 \cdot 10^{-3}$ | $6 \cdot 10^{-1}$ |
| drift corr. of $m/z = 36$ | $1 \cdot 10^{-5}$ | $4 \cdot 10^{-3}$ |
| raw $m/z = 32$ on-peak signal | $6 \cdot 10^{-4}$ | 2 |
| left $m/z = 32$ background | $-5 \cdot 10^{-6}$ | $2 \cdot 10^{-2}$ |
| right $m/z = 32$ background | $-4 \cdot 10^{-6}$ | $3 \cdot 10^{-3}$ |
| drift corr. of $m/z = 32$ | $3 \cdot 10^{-7}$ | $5 \cdot 10^{-4}$ |
| 33/32 | $2 \cdot 10^{-7}$ | $2 \cdot 10^{-4}$ |
| 34/32 | $3 \cdot 10^{-3}$ | 3 |

how the 36/32 SA ratio was calculated, the fixed curvature corrections of the $m/z = 36$ and $m/z = 32$ are denoted as $curv_1$ and $curv_2$, respectively; the latter term is set to zero, though. The stochastic 36/32 ratio was computed from measured 33/32 and 34/32 ratios (see Appendix A).

The results of the aforementioned simulations are shown in Table 7 and indicate that the minor signal, as well as its right background are among the components that influence the uncertainty of the capital delta value the most. This aligns with the results obtained for $\Delta_{35}$. Regarding relative deviations from the average, the fixed curvature correction of the $m/z = 36$ signal and the 34/32 ratio seem to be important as well. In contrast, the influence of the left $m/z = 36$ background on the relative

deviation from the measured average was even less dominant than the influence of the $m/z = 32$ signal. The components resulting in the largest relative standard deviations of $\Delta_{36}$ were the $m/z = 36$ signal (19 %), the left (6 %) and right (12 %) $m/z = 36$ backgrounds, the fixed curvature correction of the $m/z = 36$ signal (0.6 %), the $m/z = 32$ signal (2 %) and the 34/32 ratio (3 %).

It is important to note that we estimated the uncertainty of the fixed curvature correction to be around $1 \cdot 10^{-14}$ A, which

Using the measured uncertainties as standard deviations of the normal distributions, the Monte Carlo simulations show that

the largest relative deviations from the measured $\Delta_{36}$ average are associated with the fixed curvature correction (0.03 %), the right (-0.03 %), as well as left (-0.01 %) $m/z = 36$ background and the $m/z = 36$ signal (0.02 %); all of the other contributions yielded values lower than $2 \cdot 10^{-3}$ %. The relative standard deviations of $\Delta_{36}$ induced by the dominant contributors ranged from 17 % (left background of $m/z = 36$) to 51 % (raw $m/z = 36$ signal).

It is important to note that we estimated the uncertainty of the fixed curvature correction to be around $1 \cdot 10^{-14}$ A, which

is a rather conservative assumption. For instance, if the uncertainty were $5.7 \cdot 10^{-14}$ A (standard deviation of differences





between linear and polynomial fits determined during pressure-decrease measurement) instead, the relative deviation from the measured $\Delta_{36}$ value would be around -0.1 %. Consequently, the largest uncertainty would be associated with the fixed curvature correction.

## 4.5 Evolution of background

As depicted in Fig. 7, the background changes over time. Thus, it is essential to estimate the frequency at which the correlations used for the PBL corrections must be recalculated. In Table 8, we show how the averages of the clumped-isotope ratios, delta values and capital delta values vary when the same data set is corrected using PBL corrections deduced from different correlations. Specifically, we corrected 10 individual SA-ST measurements of pure-oxygen gas and performed pressure-decrease measurements (series of AV scans) on four different days within a period of two months to obtain the different correlations.

From the values listed in Table 8, it can be seen that the standard deviations of 35/32 and $\Delta_{35}$ are around 3 % and 6 %, respectively, when expressed relative to the average. In contrast, the standard deviation of the four $\delta_{35}$ values is around 21 %. When only considering the first two correlations (period of approximately 20 days), the variability of $\delta_{35}$ reduces to approximately 2 %. The standard deviations of slopes and intercepts of linear fits computed for correlations involving $m/z = 35$ signals are in the range of 0.9 % to 164 % (intercept for right background). For correlations associated with $m/z = 32$, these standard

deviations are in the range of 0.08 % to 30 % (slope for right background).

    When repeating the analysis for 36/32, it is striking that the standard deviations of slopes and intercepts of correlations involving $m/z = 36$ range from 0.2 % to 49 %, when expressed relative to the corresponding averages. Consequently, the relative standard deviations of 36/32 and its delta values were also higher, ranging from 11 % to 170 %. Excluding the last measurement, the latter value reduces to 81 %. These uncertainties are markedly larger than those related to 35/32 parameters,

though. We assume that the main cause is the curvature of the peak top, which adds a further source of uncertainty.

## 4.6 Alternative approach for pressure-decrease measurements

We also investigated whether the pressure-decrease measurements, from which the correlations are deduced, could be performed more quickly by compressing the bellows instead of waiting for the signal to decline. From Fig. 14, it can be seen that the two approaches may lead to slightly different correlations. As can be seen from Table 9, this can in turn result in significant

differences when correcting raw data. To determine the cause of this discrepancy, further studies need to be performed, which is outside the scope of this work. For all analyses presented in this paper, we used the slow measurement routine because we consider it to be more stable.

## 4.7 Correction of major oxygen isotope ratios and delta values

For certain high-precision stable-isotope studies, pressure-baseline-corrected 33/32 and 34/32 ratios might also be of inter-

est. Yeung (2018) showed that PBL corrections can indeed influence $\delta_{17}$, $\delta_{18}$ and the corresponding capital delta values. To determine if this also applies to our data, we assessed the impact of PBL corrections on 33/32, 34/32 and their delta val-



**Table 8.** Averages of PBL-corrected clumped-isotope ratios, delta values and capital delta values, deduced from 10 consecutive SA-ST measurements ($U_{av} = 4455$ V) of pure-oxygen gas (SA cylinder SC 540546 and ST cylinder SC 62349). The gas was admitted to the Elementar isoprime precisION through the NIS-II. The measurement series was performed on November 28, 2023, at a container pressure of $20.0 \pm 0.1$ mbar, an electron suppressor voltage of -140 V and an $m/z = 32$ signal around $8.3 \cdot 10^{-8}$ A. Per measurement, 13 ST intervals and 12 SA intervals were recorded (20 s integration time). The linear PBL correction was performed as described in Sect. 4.1 without applying drift corrections or corrections regarding peak top curvature. The data in the second to fourth columns were PBL-corrected using correlations determined on different days (dates indicated in the table). For comparability, for all of the corrections the background signals were determined at the same two positions. The last two columns display the average and the standard deviation of the four averages.

| Parameter | October 20, 2023 | November 10, 2023 | November 29, 2023 | January 26, 2024 | Average | Std. dev. |
|---|---|---|---|---|---|---|
| 35/32 SA [ ] | $1.51 \cdot 10^{-6}$ | $1.33 \cdot 10^{-6}$ | $1.40 \cdot 10^{-6}$ | $1.49 \cdot 10^{-6}$ | $1.43 \cdot 10^{-6}$ | $8 \cdot 10^{-8}$ |
| $\delta_{35}$ [‰] | -0.19 | -0.29 | -0.30 | -0.23 | -0.25 | 0.05 |
| $\Delta_{35}$ [‰] | -1.89 | -1.99 | -2.00 | -1.93 | -1.95 | 0.05 |
| 36/32 SA [ ] | $3.2 \cdot 10^{-6}$ | $2.5 \cdot 10^{-6}$ | $2.3 \cdot 10^{-6}$ | $2.7 \cdot 10^{-6}$ | $2.7 \cdot 10^{-6}$ | $4 \cdot 10^{-7}$ |
| $\delta_{36}$ [‰] | -1.1 | -0.2 | -0.5 | 0.3 | -3.6 | 0.6 |
| $\Delta_{36}$ [‰] | -3.3 | -2.3 | -2.6 | -1.8 | -2.5 | 0.6 |

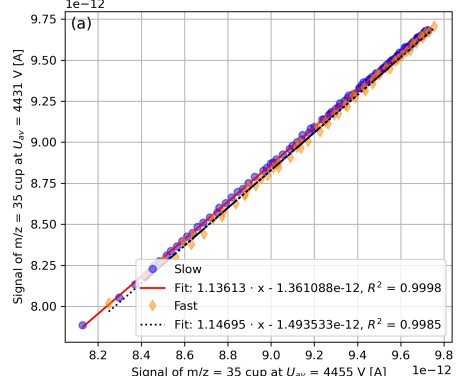
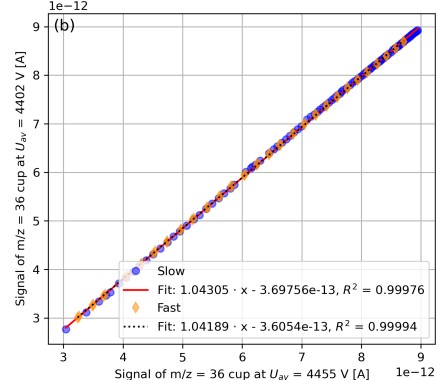

**Figure 14.** Correlation between (a) $m/z = 35$ evaluated at $U_{av} = 4455$ V (on-peak signal) and $m/z = 35$ evaluated at $U_{av} = 4431$ V (background left of peak), as well as correlation between (b) $m/z = 36$ evaluated at $U_{av} = 4455$ V and $m/z = 36$ evaluated at $U_{av} = 4402$ V (background left of peak). The correlations were determined using a series of acceleration voltage scans of pure-oxygen gas (cylinder SC 540546) admitted through the iso DUAL INLET. For the series denoted as „slow", the gas was filled into the reference bellow and then measurements were performed every 30 min. Due to the steady consumption of gas, the signal gradually decreased. For the „fast" series, the gas was filled into the sample bellow and then the signal intensity was manually varied through the compression of the bellow. In general, there was no significant idle time between the pressure adjustments and the measurements.



**Table 9.** Comparison of oxygen isotope ratios, delta values and capital delta values that corrected using linear pressure baseline corrections deduced from two different correlations (see Sect. 4.1 for the procedure). The correlations denoted as „slow" were determined through a regular pressure-decrease measurement (see Sect. 3.2). In contrast, the correlations referred to as „fast" were obtained by filling gas into the sample bellow of the iso DUAL INLET and manually altering the signal through the compression of the bellow. The corresponding acceleration voltage scans were performed at $m/z = 32$ signal intensities between $2 \cdot 10^{-8}$ A and $9 \cdot 10^{-8}$ A in steps of $2.5 \cdot 10^{-9}$ A. The idle time between measurements performed at different pressure levels was less than 3 min. The fast and the slow measurement series were performed consecutively and the data to which the corrections were applied are the same as those used for Table 8.

| Parameter | Slow | Fast |
|---|---|---|
| 35/32 SA [ ] | $1.49 \cdot 10^{-6} \pm 1 \cdot 10^{-8}$ | $1.75 \cdot 10^{-6} \pm 1 \cdot 10^{-8}$ |
| $\delta_{35}$ [‰] | $-0.2 \pm 0.2$ | $-0.9 \pm 0.2$ |
| $\Delta_{35}$ [‰] | $-2.0 \pm 0.5$ | $-2.6 \pm 0.5$ |
| 36/32 SA [ ] | $2.663 \cdot 10^{-6} \pm 6 \cdot 10^{-9}$ | $2.562 \cdot 10^{-6} \pm 5 \cdot 10^{-9}$ |
| $\delta_{36}$ [‰] | $0.3 \pm 0.1$ | $-0.3 \pm 0.1$ |
| $\Delta_{36}$ [‰] | $-1.8 \pm 0.3$ | $-2.4 \pm 0.3$ |

ues using the same 10 SA-ST measurements of pure-oxygen gas as before and applying the PBL corrections as described in Sect. 4.1. In this study, we investigated three cases: one in which all signals are pressure-baseline-corrected, a second in which only the minor signal is pressure-baseline-corrected (collector-zero-corrected major signal) and a third in which all signals are collector-zero-corrected.

The values listed in Table 10 suggest that the different corrections lead to similar measurement precisions for all measurement parameters. However, for the data set at hand, the collector-zero-corrected and PBL-corrected averages of 33/32 and 34/32 are significantly different. The reason for this discrepancy is that the collector zero values and the corresponding PBL corrections are markedly different. For $m/z = 32$, $m/z = 33$ and $m/z = 34$, the collector zero values we subtracted are approximately $1.005 \cdot 10^{-9}$, $1.008 \cdot 10^{-11}$ and $1.003 \cdot 10^{-11}$, respectively, whereas the averages of the PBL corrections applied to the SA and ST intervals of these signals are roughly $1.007 \cdot 10^{-9}$, $9 \cdot 10^{-12}$ and $4 \cdot 10^{-12}$, respectively.

We are convinced that pressure baseline corrections provide more accurate results because they estimate the background in the presence of the analyte. Moreover, a comparison of collector zeros determined on different days indicates that their variation is too small to explain the discrepancy between these values and the corresponding PBL corrections.

## 5 Influence of measurement position

As stated in Sect. 2, the measurement position can be determined through an AV scan. Based on this scan, IonOS (version 4.5) autonomously determines the centre of the peak on a predefined cup and defines the corresponding value as the new measurement position. This technique has two drawbacks. First, the peak used to determine the measurement position must be





**Table 10.** Averages of background-corrected 33/32, 34/32, $\delta_{33}$ and $\delta_{34}$ values inferred from 10 consecutive SA-ST measurements ($U_{av} = 4455$ V) of pure-oxygen gas (for measurement procedure see Table 8). The ratios were corrected through different methods: exclusively using collector zero values („coll. zeros"), exclusively using pressure baseline corrections („numerator PBL, denominator coll. zero") and applying a hybrid correction („numerator and denominator PBL"). The latter approach consisted in the application of PBL corrections to the minor signals ($m/z = 33$ and $m/z = 34$) and collector zero corrections to the $m/z = 32$ signals. The PBL corrections were computed as described in Sect. 4.1 and the indicated uncertainties correspond to the standard deviations of all independent measurement intervals of the entire measurement series (130 values for ST ratios and 120 values for SA ratios, as well as delta values). The values of the collector zeros and PBL corrections applied to the data are indicated in Sect. 4.7.

| Correction | Coll. zeros | Numerator PBL, denominator coll. zero | Numerator and denominator PBL |
|---|---|---|---|
| 33/32 SA [ ] | $7.571 \cdot 10^{-4} \pm 2 \cdot 10^{-7}$ | $7.740 \cdot 10^{-4} \pm 2 \cdot 10^{-7}$ | $7.740 \cdot 10^{-4} \pm 2 \cdot 10^{-7}$ |
| 33/32 ST [ ] | $7.566 \cdot 10^{-4} \pm 2 \cdot 10^{-7}$ | $7.736 \cdot 10^{-4} \pm 2 \cdot 10^{-7}$ | $7.736 \cdot 10^{-4} \pm 2 \cdot 10^{-7}$ |
| $\delta_{33}$ [‰] | $0.62 \pm 0.02$ | $0.59 \pm 0.02$ | $0.59 \pm 0.02$ |
| 34/32 SA [ ] | $4.0677 \cdot 10^{-3} \pm 6 \cdot 10^{-7}$ | $4.1446 \cdot 10^{-3} \pm 7 \cdot 10^{-7}$ | $4.1447 \cdot 10^{-3} \pm 7 \cdot 10^{-7}$ |
| 34/32 ST [ ] | $4.0633 \cdot 10^{-3} \pm 7 \cdot 10^{-7}$ | $4.1401 \cdot 10^{-3} \pm 7 \cdot 10^{-7}$ | $4.1402 \cdot 10^{-3} \pm 7 \cdot 10^{-7}$ |
| $\delta_{34}$ [‰] | $1.08 \pm 0.05$ | $1.08 \pm 0.05$ | $1.08 \pm 0.05$ |

a dominant peak; if the peak is too small, the aforementioned version of IonOS cannot detect it. Second, if the Faraday cups are
not perfectly aligned, the centres of the peaks recorded on different cups are located at slightly different acceleration voltages. Therefore, it is advisable to choose a narrow cup for determining the measurement position.

In Fig. 15, we present scaled versions of the five oxygen peaks measured in pure-oxygen gas (cylinder SC 62349), from which the relative positioning of the cups can be inferred. From this figure it can also be seen that the cups of our instrument are not perfectly aligned and that the clumped-isotope peaks substantially vary over the peak width.

To study the influence of the measurement position on the oxygen isotope ratios and their delta values, we performed 10 SA-ST measurements (SA cylinder SC 540546 and ST cylinder SC 62349) at three different acceleration voltages: 4450 V, 4455 V and 4465 V. To assess the reproducibility of our results, we repeated the entire measurement series one week after conducting the first one.

The results of this study, which are shown in Table 11, indicate a considerable dependence of the isotope ratios' uncertainty
on the measurement position. In general, for both measurement runs, the standard deviation of the isotope ratios was lowest at 4450 V and highest at 4455 V. The averages of the isotope ratios computed over all measurement runs and measurement positions suggest that the most stable ratios are 33/32 and 34/32; the standard deviations relative to the corresponding averages are around 0.3 % and 0.06 %, respectively. In contrast, for 35/32 and 36/32, these values are around 5 % and 13 %, respectively. Additionally, $\delta_{33}$ proved to be slightly more stable than $\delta_{34}$. The relative standard deviation of the $\delta_{35}$ and $\delta_{36}$ is roughly 26 %
and 74 %, respectively. Repeating the computations for $\Delta_{35}$ and $\Delta_{36}$ shows that the variations are similar, namely around 13 %



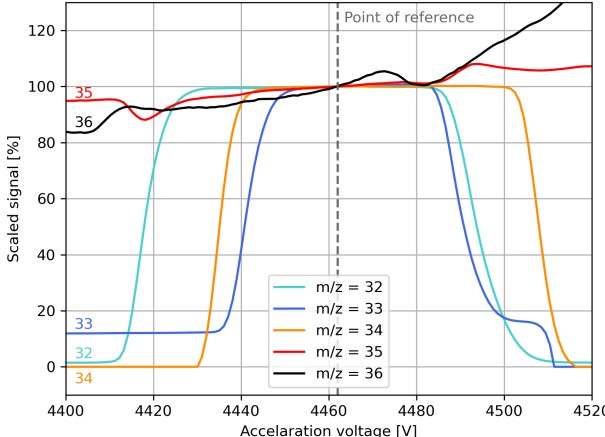

**Figure 15.** Scaled oxygen peaks recorded during an acceleration voltage scan of pure-oxygen gas (cylinder SC 62349). The scaling was performed with respect to the point of reference indicated in the plot. All depicted signals were recorded simultaneously on different cups.

and 12 %, respectively. Generally, these findings are directly related to the peak shapes – the more variable the peak relative to its height and width, the less stable the measurement parameters associated with this peak.

Regarding the reproducibility of delta and capital delta averages, most variations fell within the measurement uncertainties. While the standard deviations of $\Delta_{35}$ and $\Delta_{36}$ are similar at 4450 V and 4465 V, they are a factor of 2 to 7 higher at 4455 V. However, it is important to note that for other measurement series conducted at 4455 V, uncertainties as low as those obtained at 4450 V and 4465 V were achieved (e.g., see Table 6). This suggests that the observed discrepancies may also be influenced by differences in the quality of the pressure baseline corrections applied to the clumped-isotope signals.

It is important to note that the SA-ST measurements conducted for the studies presented in previous sections were performed prior to this study. As mentioned in Sect. 2, the acceleration voltage was usually set to 4455 V because it is close to the centre of the $m/z = 35$ peak, while 4465 V is near the right edge of the $m/z = 36$ peak and 4450 V is close to the left edges of the $m/z = 33$, as well as $m/z = 34$ peaks. The results of this section suggest that for certain measurement parameters higher precisions might be obtained at 4450 V or 4465 V, though.

Comparing the delta and capital delta values listed in Table 11 to the corresponding values presented in previous sections reveals noticeable differences in certain absolute values. This is an effect resulting from improperly calibrated isotope ratios, which is best illustrated when comparing 35/32 averages. For instance, accounting for the difference between the 35/32 averages used in Sect. 4.4.1 and the corresponding 35/32 averages of the first measurement run listed in Table 11 ($U_{av} = 4455$ V), the $\Delta_{35}$ value reported in Sect. 4.4.1 increases from approximately $-2.0$ ‰ to $-1.1$ ‰, thus agreeing with the results shown in Table 11. To obtain the corrected value, we applied a very simple correction aimed at demonstrating the main problem; we first separated SA from ST intervals, computed the average 35/32 SA ratio for each measurement series and finally added their




**Table 11.** Oxygen isotope ratios (ST) and delta values deduced from SA-ST measurements of pure-oxygen gas (SA cylinder SC 540546 and ST cylinder SC 62349) performed with the Elementar isoprime precisION and the NIS-II. The listed averages were calculated from 10 individual measurements, each consisting of 13 ST and 12 SA intervals (20 s integration). Each series was conducted at three different acceleration voltages (indicated at the top of the last three columns) and was repeated after one week (indicated as 1 and 2). The $m/z = 32$ signal intensities were all approximately $6.3 \cdot 10^{-8} \pm 1 \cdot 10^{-9}$ A. The uncertainties correspond to the standard deviations of all independent interval means recorded during the measurement series (130 isotope ratios and 120 delta values). Additionally, for each measurement, one value for $\Delta_{35}$ and $\Delta_{36}$ was calculated (for the procedure see Sect. 4.4.1). The SA and ST ratios involved in these computations were fixed for each series (10 measurements) individually. The uncertainties of the capital delta averages correspond to the standard deviations of the 10 values. The $m/z = 32$, $m/z = 33$ and $m/z = 34$ signals were collector-zero-corrected, while the remaining signals were corrected using linear PBL corrections (see Sect. 4.1). For both measurement runs, the correlations used for the PBL correction were calculated from separate series of AV scans recorded one to two days prior to the start of the corresponding run.

| Parameter | Measurement run | 4450 V | 4455 V | 4465 V |
|---|---|---|---|---|
| 33/32 [ ] | 1 | $7.5357 \cdot 10^{-4} \pm 8 \cdot 10^{-8}$ | $7.570 \cdot 10^{-4} \pm 4 \cdot 10^{-7}$ | $7.582 \cdot 10^{-4} \pm 1 \cdot 10^{-7}$ |
| 33/32 [ ] | 2 | $7.5327 \cdot 10^{-4} \pm 8 \cdot 10^{-8}$ | $7.560 \cdot 10^{-4} \pm 2 \cdot 10^{-7}$ | $7.579 \cdot 10^{-4} \pm 1 \cdot 10^{-7}$ |
| 34/32 [ ] | 1 | $4.0688 \cdot 10^{-3} \pm 2 \cdot 10^{-7}$ | $4.075 \cdot 10^{-3} \pm 1 \cdot 10^{-6}$ | $4.0714 \cdot 10^{-3} \pm 5 \cdot 10^{-7}$ |
| 34/32 [ ] | 2 | $4.0700 \cdot 10^{-3} \pm 3 \cdot 10^{-7}$ | $4.0740 \cdot 10^{-3} \pm 9 \cdot 10^{-7}$ | $4.0724 \cdot 10^{-3} \pm 5 \cdot 10^{-7}$ |
| 35/32 [ ] | 1 | $1.496 \cdot 10^{-6} \pm 4 \cdot 10^{-9}$ | $1.33 \cdot 10^{-6} \pm 2 \cdot 10^{-8}$ | $1.357 \cdot 10^{-6} \pm 6 \cdot 10^{-9}$ |
| 35/32 [ ] | 2 | $1.515 \cdot 10^{-6} \pm 5 \cdot 10^{-9}$ | $1.40 \cdot 10^{-6} \pm 1 \cdot 10^{-8}$ | $1.437 \cdot 10^{-6} \pm 7 \cdot 10^{-9}$ |
| 36/32 [ ] | 1 | $2.596 \cdot 10^{-6} \pm 2 \cdot 10^{-9}$ | $2.741 \cdot 10^{-6} \pm 9 \cdot 10^{-9}$ | $3.312 \cdot 10^{-6} \pm 3 \cdot 10^{-9}$ |
| 36/32 [ ] | 2 | $2.341 \cdot 10^{-6} \pm 4 \cdot 10^{-9}$ | $2.534 \cdot 10^{-6} \pm 6 \cdot 10^{-9}$ | $3.118 \cdot 10^{-6} \pm 3 \cdot 10^{-9}$ |
| $\delta_{33}$ [‰] | 1 | $0.58 \pm 0.02$ | $0.58 \pm 0.02$ | $0.58 \pm 0.02$ |
| $\delta_{33}$ [‰] | 2 | $0.58 \pm 0.02$ | $0.58 \pm 0.02$ | $0.58 \pm 0.02$ |
| $\delta_{34}$ [‰] | 1 | $1.07 \pm 0.03$ | $1.07 \pm 0.04$ | $1.08 \pm 0.03$ |
| $\delta_{34}$ [‰] | 2 | $1.08 \pm 0.03$ | $1.08 \pm 0.03$ | $1.08 \pm 0.03$ |
| $\delta_{35}$ [‰] | 1 | $0.7 \pm 0.5$ | $0.5 \pm 0.5$ | $0.4 \pm 0.4$ |
| $\delta_{35}$ [‰] | 2 | $0.4 \pm 0.5$ | $0.4 \pm 0.4$ | $0.4 \pm 0.3$ |
| $\delta_{36}$ [‰] | 1 | $0.2 \pm 0.2$ | $0.1 \pm 0.2$ | $0.1 \pm 0.2$ |
| $\delta_{36}$ [‰] | 2 | $0.7 \pm 0.5$ | $0.6 \pm 0.4$ | $0.4 \pm 0.3$ |
| $\Delta_{35}$ [‰] | 1 | $-0.9 \pm 0.2$ | $-1.1 \pm 0.9$ | $-1.3 \pm 0.3$ |
| $\Delta_{35}$ [‰] | 2 | $-1.2 \pm 0.2$ | $-1.2 \pm 0.6$ | $-1.3 \pm 0.2$ |
| $\Delta_{36}$ [‰] | 1 | $-1.9 \pm 0.1$ | $-2.0 \pm 0.7$ | $-2.0 \pm 0.2$ |
| $\Delta_{36}$ [‰] | 2 | $-1.5 \pm 0.2$ | $-1.6 \pm 0.5$ | $-1.8 \pm 0.3$ |

difference to the data used in Sect. 4.4.1. Before re-calculating the $\Delta_{35}$ values, the procedure was repeated for the ST ratios, which were corrected accordingly.



Given the relatively large uncertainties associated with our pressure baseline corrections, it is vital to account for deviations
from the expected values, e.g., by measuring gases with known isotopic composition or gases conforming to the stochastic
distribution. Furthermore, the capital delta values depend on the stochastic isotope ratios involved in their calculation. Thus,
it is also necessary to monitor these ratios and adjust the capital delta values accordingly. Although such corrections are
important, they are outside the scope of this study, whose main purpose was to determine an approach for correcting peaks
with non-square shapes. In clumped-isotope studies of $CO_2$, inter-laboratory comparison has been an issue for several years.
Probably the most widely adopted approach is the „Absolute Reference Frame" introduced by Dennis (2011) directly tying the
capital delta values to the theoretical equilibrium values calculated by Wang (2004).

## 6 Conclusions

In this study, we demonstrated that our $m/z = 35$ and $m/z = 36$ signals, measured in pure-oxygen gas using a low-mass-
resolution IRMS, have very small signal-to-baseline ratios (1.005 and 1.025, respectively) and that their peaks substantially
deviate from the expected square shape. In general, the $m/z = 36$ peak tops exhibit a marked negative curvature, while the
$m/z = 35$ peak tops increase linearly with the acceleration voltage. Furthermore, we showed that applying an electron sup-
pressor voltage of -140 V is insufficient to obtain positive $m/z = 35$ and $m/z = 36$ signals if collector zero values are used for
background correction. However, we observed that lowering the electron suppressor voltage and decreasing the source pres-
sure both lead to an increase in the clumped-isotope signals. In accordance with the study by Bernasconi (2013), we concluded
that the pressure-dependent background is significantly affected by secondary electrons. In addition, we noticed that the peak
shapes evolve over time and that not only does the magnitude of the peak change with signal intensity, but also its shape.
Consequently, predicting the peak shape at a specific signal intensity based on a single acceleration voltage scan is difficult.
We assume that the variation in peak shape mainly arises from changes in the background.

Moreover, we extensively discussed our routine for determining pressure baseline corrections, which estimate the back-
ground in the presence of the analyte. Essentially, our approach differs from the PBL corrections presented by Bernasconi
(2013) in that we use the on-peak signal of a given peak to predict its background signals and in that we compute not only
one correlation for the determination of the minimum background but also two or more correlations to account for the slope
and/or curvature of the peak tops. This method allows for more accurate modelling of peak shapes and considers differences in
off-peak signals left and right of the peaks.

The main part of our work was dedicated to studies aimed at determining and improving our PBL corrections. For each
of these studies, we presented evaluations of different measurement parameters (isotope ratios, delta values and capital delta
values) related to our $m/z = 35$ and $m/z = 36$ signals. Our main findings are as follows:

– Our PBL corrections significantly reduce the influence of secondary electrons on clumped-isotope signals of oxygen,
resulting in an increase (instead of decrease) with source pressure, which follows the trend of the $m/z = 32$ signals (see
Fig. 10).



- The background of a given peak is best predicted using its on-peak signals rather than signals recorded on adjacent cups. The coefficients of determination of the linear fits computed for the corresponding correlations differ at the $10^{-3}$ and $10^{-2}$ levels for $m/z = 35$ and $m/z = 36$, respectively. The impact on the precision is appreciable, though. For instance, using on-peak signals of $m/z = 35$ instead of $m/z = 40$ (the next best correlation) halves the uncertainty of 35/32. Nonetheless, the uncertainty of $\delta_{35}$ remains similar (see Table 2).

- Correcting raw signals by subtracting only the off-peak signal from either side of the corresponding peak may lead to considerable underestimation or overestimation of the actual signal, which can impact accuracy and precision. Based on 35/32 and $\delta_{35}$, we showed that certain values may be unrealistic if single values are subtracted from the raw signals – even if the ratios are all positive. Subtracting an off-peak signal to the right of the $m/z = 35$ peak from the raw signal resulted in 35/32 averages and uncertainties that were approximately 1 order of magnitude off, yielding delta values that deviated by hundreds to thousands of permil (see Table 3). Hence, for our clumped-isotope signals, it is essential to account for the background on both sides of the peak. Additionally, to obtain 36/32 ratios within the expected range, the curvature of $m/z = 36$ peak tops has to be considered when applying PBL corrections, e.g., by using a polynomial instead of a linear PBL correction.

- We demonstrated that markedly higher precisions can be obtained when PBL corrections are applied to individual measurement intervals rather than using the same value for all intervals. This is because intensity variations are taken into account. Table 3 shows that sub-permil uncertainties of $\delta_{35}$ could only be achieved with this approach. For 35/32, the uncertainties improved by approximately 1 order of magnitude. Correcting individual intervals of $m/z = 35$ signals and linearly interpolating off-peak signals to the left and right of the peak yielded 35/32 uncertainties around $1 \cdot 10^{-8}$ instead of $2 \cdot 10^{-8}$ (correction of individual intervals without interpolation). Both methods yielded an uncertainty of $0.2\,‰$ for $\delta_{35}$, though (see Table 3).

- Applying drift corrections to PBL-corrected isotope ratios can further improve measurement precision. This is evident from Table 4, which shows that the application of a linear drift correction to 35/32 ratios reduces the uncertainty by 1 order of magnitude and that a polynomial drift correction provides additional improvement, albeit to a lesser extent (uncertainty around $8 \cdot 10^{-10}$ instead of $1 \cdot 10^{-9}$). For 36/32, we observed a difference of up to a factor of 7 (see Table 6). Within the measurement uncertainty, the drift corrections left the averages and uncertainties of our delta and capital delta values unchanged (see Table 4 and Table 6).

- The application of pressure baseline corrections to raw measurement signals resulted in standard deviations of approximately $1 \cdot 10^{-9}$ (35/32), $0.2\,‰$ ($\delta_{35}$), $0.5\,‰$ ($\Delta_{35}$), $7 \cdot 10^{-9}$ (36/32), $0.1\,‰$ ($\delta_{36}$) and $0.1\,‰$ ($\Delta_{36}$) for at least 120 intervals (total analysis and integration time around 6 h and 40 min, respectively). For $\Delta_{35}$ and $\Delta_{36}$, the corresponding standard errors of the mean are less than $0.05\,‰$ and $0.01\,‰$, respectively. The uncertainties of certain measurement parameters could be further reduced by optimising the measurement position (acceleration voltage) and applying addi-





tional drift corrections. However, due to uncertainties associated with the correction of the $m/z = 36$ peak top curvature, the uncertainties of $\delta_{36}$ and $\Delta_{36}$ might actually be a few tenths of permil higher than the indicated $0.1\,‰$.

685     – It is vital to study the influence of measurement position on the results, as precision and accuracy can vary significantly, even for major signals. For instance, at an acceleration voltage of 4450 V, the standard deviation for 34/32 was approximately $2 \cdot 10^{-7}$ to $3 \cdot 10^{-7}$, whereas at 4455 V, it ranged from $9 \cdot 10^{-7}$ to $1 \cdot 10^{-6}$ (see Table 11).

    – An apparent dependence of the capital delta values on the corresponding delta values (non-linearity) can arise from differences in the precision of the isotope ratios and their stochastic counterparts (see Fig. 11) used in the calculation of 690     the capital delta values.

    – Monte Carlo simulations indicated that the uncertainties of $\Delta_{35}$ and $\Delta_{36}$ are primarily driven by the minor on-peak signals and their predicted background values (see Table 5 and Table 7). For $\Delta_{36}$, the uncertainty associated with the curvature of the $m/z = 36$ peak top is also substantial. This was also noted in Sect. 4.4.4, where we showed that polynomial PBL corrections can alter the peak height, mainly affecting the accuracy of the measurement results (see Table 6).

695     – We also applied PBL corrections to $m/z = 32$, $m/z = 33$ and $m/z = 34$ signals, but did not observe any noteworthy discrepancies between the uncertainties of PBL-corrected and collector-zero-corrected 33/32, 34/32, $\delta_{33}$ and $\delta_{34}$ values. Conversely, significant differences in the absolute values of 33/32 and 34/32 were obtained (see Table 10).

While the focus of our study was primarily on the precision of isotope ratios, delta values and capital delta values associated with $m/z = 35$ and $m/z = 36$ signals, the calibration of their absolute values remains an issue. Our data still lack such cali- 700 bration, which is necessary for enabling inter-laboratory comparisons and for comparing our absolute values recorded during different measurement series. Determining such corrections is part of our ongoing work, which also includes enhancing the accuracy of corrections for curved peak tops.

*Code and data availability.*   The data and code are both available upon request (stephan.raess@unibe.ch).

## Appendix A: Calculation of stochastic ratios

705 The stochastic values for the isotope ratios 35/32 ($^{35}R$) and 36/32 ($^{36}R$) can be calculated from the measured 33/32 ($^{33}R$) and 34/32 ($^{34}R$) ratios as follows:

$$^{33}R = 2 \cdot {}^{17}R \;\; \Rightarrow \;\; {}^{17}R = \frac{^{33}R}{2} \tag{A1}$$

$$^{34}R = \left(^{17}R\right)^2 + 2 \cdot {}^{18}R \;\; \Rightarrow \;\; {}^{18}R = \frac{^{34}R - \left(^{17}R\right)^2}{2} \tag{A2}$$



$$^{35}R \;=\; 2 \,\cdot\, {}^{17}R \,\cdot\, {}^{18}R \;=\; {}^{33}R\;\frac{{}^{34}R - \left(\frac{{}^{33}R}{2}\right)^2}{2} \tag{A3}$$

$$^{36}R \;=\; ({}^{18}R)^2 \;=\; \left(\frac{{}^{34}R - \left(\frac{{}^{33}R}{2}\right)^2}{2}\right)^2 \tag{A4}$$

To express 35/32 and 36/32 as functions of 33/32 and 34/32 only, we substituted the results of Eq. (A1) and Eq. (A2) into Eq. (A3) and Eq. (A4). The variables $^{17}R$ and $^{18}R$ denote the elemental oxygen isotope ratios 17/16 and 18/16, respectively.

## Appendix B: Uncertainty of capital delta value

To estimate the uncertainty of the capital delta value, we apply the propagation of uncertainty to Eq. (1):

$$u\left(\Delta_A(\permil)\right) = \Bigg[\left(\frac{1000}{{}^{A}R^*} \cdot u({}^{A}R)\right)^2 + \left(-1000\frac{{}^{A}R}{{}^{A}R^{*2}} \cdot u({}^{A}R^*)\right)^2$$
$$- 2\cdot\frac{{}^{A}R\cdot 10^6}{{}^{A}R^{*3}} \cdot u({}^{A}R)\cdot u({}^{A}R^*)\cdot\rho({}^{A}R,{}^{A}R^*)\Bigg]^{\frac{1}{2}}\permil \tag{B1}$$

In Eq. (B1), $u$ denotes the uncertainty (standard deviation) and $\rho({}^{A}R,{}^{A}R^*)$ represents the correlation coefficient associated with $^{A}R$ and $^{A}R^*$. In this section, correlation terms are always included because not all variables involved in our calculations are independent of one another. The pressure-baseline-corrected isotope ratio of the sample gas can be expressed as follows:

$$^{A}R = \frac{S_{1,raw} - w_{1,left}\cdot S_{1,bg,left} - w_{1,right}\cdot S_{1,bg,right} - d_1 + curv_1}{S_{2,raw} - w_{2,left}\cdot S_{2,bg,left} - w_{2,right}\cdot S_{2,bg,right} - d_2 + curv_2}. \tag{B2}$$

In Eq. (B2), $S_{i,raw}$ represents the raw measurement signal and $d_i$ denotes the corresponding drift corrections. The variables $S_{i,bg,left}$ and $S_{i,bg,right}$ correspond to the estimated background signals left and right of the peak (see Sect. 4.1), respectively. Since the background correction at the measurement position is obtained by interpolating the left and right background signals (see Sect. 4.1), we introduced the weights $w_{i,left}$ and $w_{i,right}$ (i.e., if the measurement position is exactly between the left and right background position, the weights are equal to 0.5). If the peak tops are curved (e.g., $m/z = 36$ signal shown in Fig. 13), the linear background correction is not appropriate and has to be adjusted. To account for this difference, we introduced the $curv_i$ terms. It is important to emphasize that this is merely a simple correction, enabling us to assess the influence of the BG terms and the curvature on the uncertainty independently. For correcting $m/z = 36$ signals recorded during SA-ST measurements, we typically apply polynomial PBL corrections (see Sect. 4.4.4).

Denoting the correlation coefficient associated with the $i$-th and the $j$-th components as $\rho_{i,j}$, the propagation of uncertainty yields





$$u\left({}^{A}R\right) = \sqrt{\sum_{i=1}^{6} c_i^2 + 2 \cdot \sum_{i=1}^{5} \sum_{j=i+1}^{6} c_i \cdot c_j \cdot \rho_{i,j}},$$

(B3)

for the uncertainty of ${}^{A}R$, given that

a) no curvature correction is applied

b) the uncertainties associated with the weights are neglected

c) $c_1 \doteq \frac{1}{S_{2,raw}-w_{2l}\cdot S_{2,bg,left}-w_{2,right}\cdot S_{2,bg,right}-d_2} \cdot u(S_{1,raw})$

d) $c_2 \doteq -\frac{w_{1,left}}{S_{2,raw}-w_{2,left}\cdot S_{2,bg,left}-w_{2,right}\cdot S_{2,bg,right}-d_2} \cdot u(S_{1,bg,left})$

e) $c_3 \doteq -\frac{w_{1,right}}{S_{2,raw}-w_{2,left}\cdot S_{2,bg,left}-w_{2,right}\cdot S_{2,bg,right}-d_2} \cdot u(S_{1,bg,right})$

f) $c_4 \doteq -\frac{1}{S_{2,raw}-w_{2,left}\cdot S_{2,bg,left}-w_{2,right}\cdot S_{2,bg,right}-d_2} \cdot u(d_1)$

g) $c_5 \doteq -\frac{S_{1,raw}-w_{1,left}\cdot S_{1,bg,left}-w_{1,right}\cdot S_{1,bg,right}-d_1}{(S_{2,raw}-w_{2l}\cdot S_{2,bg,left}-w_{2,right}\cdot S_{2,bg,right}-d_2)^2} \cdot u(S_{2,raw})$

h) $c_6 \doteq w_{2,left} \cdot \frac{S_{1,raw}-w_{1,left}\cdot S_{1,bg,left}-w_{1,right}\cdot S_{1,bg,right}-d_1}{(S_{2,raw}-w_{2,left}\cdot S_{2,bg,left}-w_{2,right}\cdot S_{2,bg,right}-d_2)^2} \cdot u(S_{2,bg,left})$

i) $c_7 \doteq w_{2,right} \cdot \frac{S_{1,raw}-w_{1,left}\cdot S_{1,bg,left}-w_{1,right}\cdot S_{1,bg,right}-d_1}{(S_{2,raw}-w_{2,left}\cdot S_{2,bg,left}-w_{2,right}\cdot S_{2,bg,right}-d_2)^2} \cdot u(S_{2,bg,right})$

j) $c_8 \doteq \frac{S_{1,raw}-w_{1,left}\cdot S_{1,bg,left}-w_{1,right}\cdot S_{1,bg,right}-d_1}{(S_{2,raw}-w_{2l}\cdot S_{2,bg,left}-w_{2,right}\cdot S_{2,bg,right}-d_2)^2} \cdot u(S_{d_2})$

Furthermore, neglecting drift corrections, assuming that the two background values are identical ($S_{i,bg,left} = S_{i,bg,right} = 0.5 \cdot S_{i,bg}$) and setting the weights to 0.5 (measurement position exactly between left and right background position), Eq. (B2) reduces to

$${}^{A}R = \frac{S_{1,raw} - S_{1,bg}}{S_{2,raw} - S_{2,bg}}.$$

(B4)

Applying the propagation of uncertainty to Eq. (B4) results in

$$u\left({}^{A}R\right) = \Bigg[ \left(\frac{1}{S_{2,raw}-S_{2,bg}} \cdot u(S_{1,raw})\right)^2 + \left(-\frac{1}{S_{2,raw}-S_{2,bg}} \cdot u(S_{1,bg})\right)^2$$

$$+ \left(-\frac{S_{1,raw}-S_{1,bg}}{(S_{2,raw}-S_{2,bg})^2} \cdot u(S_{2,raw})\right)^2 + \left(\frac{S_{1,raw}-S_{1,bg}}{(S_{2,raw}-S_{2,bg})^2} \cdot u(S_{2,bg})\right)^2$$

$$- 2 \cdot \frac{1}{(S_{2,raw}-S_{2,bg})^2} \cdot u(S_{1,raw}) \cdot u(S_{1,bg}) \cdot \rho(S_{1,raw}, S_{1,bg})$$

$$- 2 \cdot \frac{S_{1,raw}-S_{1,bg}}{(S_{2,raw}-S_{2,bg})^3} \cdot u(S_{1,raw}) \cdot u(S_{2,raw}) \cdot \rho(S_{1,raw}, S_{2,raw})$$

$$+ 2 \cdot \frac{S_{1,raw}-S_{1,bg}}{(S_{2,raw}-S_{2,bg})^3} \cdot u(S_{1,raw}) \cdot u(S_{2,bg}) \cdot \rho(S_{1,raw}, S_{2,bg})$$





$$+ 2 \cdot \frac{S_{1,raw} - S_{1,bg}}{(S_{2,raw} - S_{2,bg})^3} \cdot u(S_{1,bg}) \cdot u(S_{2,raw}) \cdot \rho(S_{1,bg}, S_{2,raw})$$

$$- 2 \cdot \frac{S_{1,raw} - S_{1,bg}}{(S_{2,raw} - S_{2,bg})^3} \cdot u(S_{1,bg}) \cdot u(S_{2,bg}) \cdot \rho(S_{1,bg}, S_{2,bg})$$

$$- 2 \cdot \frac{(S_{1,raw} - S_{1,bg})^2}{(S_{2,raw} - S_{2,bg})^4} \cdot u(S_{2,raw}) \cdot u(S_{2,bg}) \cdot \rho(S_{2,raw}, S_{2,bg}) \Bigg]^{\frac{1}{2}}. \tag{B5}$$

For collector-zero-corrected measurements to which no drift corrections are applied, the uncertainty can be estimated using Eq. (B5) since a single value is subtracted from the raw signals to account for the background.

Unfortunately, generalising $^A R$ and $^A R^*$ is not possible due to their dependence on the components involved in their calculation. For instance, as shown in Eq. (A3), $^{35} R^*$ can be expressed as

$$^{35} R^* = 2 \cdot {}^{17} R \cdot {}^{18} R = 2 \cdot \frac{^{33} R}{2} \cdot \frac{^{34} R - {}^{17} R^2}{2} = \frac{^{33} R \cdot {}^{34} R}{2} - \frac{^{33} R^3}{8}. \tag{B6}$$

According to the propagation of uncertainty, the standard deviation of this quantity is given by

$$u\left( {}^{35} R^* \right) = \Bigg[ \left( \left( \frac{^{34} R}{2} - \frac{3 \cdot {}^{33} R^2}{8} \right) \cdot u({}^{33} R) \right)^2 + \left( \frac{^{33} R}{2} \cdot u({}^{34} R) \right)^2$$

$$+ \left( \frac{^{33} R \cdot {}^{34} R}{2} - \frac{3 \cdot {}^{33} R^3}{8} \right) \cdot u({}^{33} R) \cdot u({}^{34} R) \cdot \rho({}^{33} R, {}^{34} R) \Bigg]^{\frac{1}{2}}. \tag{B7}$$

It is worth mentioning that the uncertainties of the isotope ratios 33/32 and 34/32 (denoted as $u({}^{33} R)$ and $u({}^{34} R)$, respectively) also have to be calculated using Eq. (B5).

Based on Eq. (A4), which can be rearranged to

$$^{36} R^* = \left( {}^{18} R \right)^2 = \left( \frac{^{34} R - {}^{17} R^2}{2} \right)^2 = \left( \frac{^{34} R}{2} - \frac{^{33} R^3}{8} \right)^2, \tag{B8}$$

the propagation of uncertainty yields

$$u({}^{36} R^*) = \Bigg[ \left( {}^{33} R^2 \cdot \frac{3}{4} \cdot \left( \frac{^{34} R}{2} - \frac{^{33} R^3}{8} \right) \cdot u({}^{33} R) \right)^2 + \left( \left( \frac{^{34} R}{2} - \frac{^{33} R^3}{8} \right) \cdot u({}^{34} R) \right)^2$$

$$+ {}^{33} R^2 \cdot \frac{3}{4} \cdot \left( \frac{^{34} R}{2} - \frac{^{33} R^3}{8} \right)^2 \cdot u({}^{33} R) \cdot u({}^{34} R) \cdot \rho({}^{33} R, {}^{34} R) \Bigg]^{\frac{1}{2}} \tag{B9}$$

for the uncertainty of $^{36} R^*$.

*Author contributions.* SR was in charge of investigation. PN and WP provided technical support. The formal analysis and validation of the gathered data were carried out by SR and ML. SR was in charge of visualisation and wrote the original draft of the paper. In the reviewing and editing process SR, ML, PN and PW were involved. For funding acquisition ML and PW were responsible.



*Competing interests.* The authors declare that they have no conflict of interest.

*Acknowledgements.* First of all, we would like to express our gratitude to Elementar Analysensysteme GmbH, Elementar-Straße 1, D-63505 Langenselbold, Germany, and in particular Elementar UK Ltd., Isoprime House, Earl Road, Cheadle Hulme, Stockport - SK8 6PT, United 780 Kingdom, which provided technical and financial support for this work. We are also grateful to the Swiss National Science Foundation (SNF-Project 172550) for their financial contributions. Furthermore, we sincerely thank the members of the workshop teams of the division of Climate and Environmental Physics of the University of Bern, whose effort regarding development and maintenance of our setup was indispensable. We also acknowledge the use of artificial intelligence for grammar and spell checking, as well as for enhancing sentence clarity and code development (Grammarly, Microsoft Copilot (GPT-4 Turbo) and ChatGPT 4.0).



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
