# Peer review of "Determination of pressure baseline corrections for clumped-isotope signals with complex peak shapes"

_EGUsphere, 2024_

## Referee Comment (RC2)

825

[referee-annotated manuscript omitted]

---

## Author Response (AR1)

**Author's response**

Below, we repeat the reviewers' comments (in black), followed by our responses (in italics, blue text).

**RC 1**

Line 78 "By definition" instead of "by construction"

*Done.*

Lines 80-100 this paragraph should be shortened and updated. It is now clear that the negative backgrounds are the cause of the observed nonlinearities and PBL corrections are a must in clumped isotope analysis. HG and EG corrections are a wourkaround that does not correct the direct cause of the nonlinearity and are unnecessarily complicated.

Many papers have shown that PBL corrections eliminates the need for HG an EG corrections. He 2012 and Bernasconi 2013 were the start, but other papers have confirmed it, and all the results in the interlaboratory calibration exercise presented in Bernasconi et al. 2021 only utilize PBL corrected data.

*The introduction was shortened and updated.*

Lines 117-123 not really necessary,could be removed.

*The corresponding lines were removed.*

Line 124: it is not necessary to hit the center of the cup, the flat portion of the signal represent the width of the cup where the entire beam is collected in the cup. The signal can also be measured correctly on the side of the peak, as long as it's in the flat part of the peak.

*The following sentences were deleted: "To collect as many ions as possible, all ion beams of interest must hit the centres of the corresponding cups, which requires an ideal geometrical alignment of the cups. However, the smaller the mass resolution of the mass spectrometer, the more challenging it becomes to achieve this goal."*

Bottom of page 9 : beginning of the sentence is missing.

*Nothing was changed because the sentence starts on the previous page. This is confusing, as there are various plots in between.*

Line 230-235 you could mention that such large changes in background shapes were also reported by Meckler et al. 2014 (DOI: 10.1002/rcm.6949) for a MAT253 mas spectrometer. As you mention, it is indeed very important to monitor backgrounds with scans as the shape

and magnitude can strongly change.  Just peak jumping to the side of the peak is not a sufficient way to monitor the change in background.

*We added the sentence "Similar observations were made by Meckler (2014) using a Thermo Fisher MAT 253 mass spectrometer."*

Line 437 what do you mean that the procedure for correcting is "more involved"?

*The sentence "While our procedure for correcting m/z=35 signals is already more involved than the PBL corrections presented by He (2012) and Bernasconi (2013), m/z=36 peaks introduce an additional layer of complexity due to their curvature." was replaced by "While accounting for off-peak signals to the left and right of the m/z = 35 peaks already complicates the calculation of PBL corrections compared to the procedure outlined by He (2012) and Bernasconi (2013), the m/z = 36 peaks introduce an additional layer of complexity due to their curvature."*

Line 580 this is only a problem on curved peak tops, it the peak top is flat than the exact position is not so important.

*The sentences "Second, if the Faraday cups are not perfectly aligned, the centres of the peaks recorded on different cups are located at slightly different acceleration voltages. Therefore, it is advisable to choose a narrow cup for determining the measurement position." were replaced by "Second, if the Faraday cups are not perfectly aligned, the centres of the peaks recorded on different cups are located at slightly different acceleration voltages, which can pose a problem when the peak tops are not flat. In this case, it is advisable to choose a narrow cup for determining the measurement position."*

625  But now it is I-CDES (Bernasconi et al. 2021), the tying to the theoretical values is the first step, but the interlaboratory comparability can only be ensured by using traceable standards that can be measured in different laboratories.

*The sentence "Probably the most widely adopted approach is the Absolute Reference Frame introduced by Dennis (2011) directly tying the capital delta values to the theoretical equilibrium values calculated by Wang (2004)." was replaced by "Probably the most widely adopted approach nowadays is the Intercarb-Carbon Dioxide Equilibrium Scale (I-CDES) introduced by Bernasconi (2021), which ties the capital delta values to the theoretical equilibrium values calculated by Wang (2004) and ensures interlaboratory comparability through the use of traceable standards."*

The code and data should be made available on a repository.

*Once this paper is published, the basic code and its documentation will be made available in the Bern Open Repository and Information System (BORIS) as part of the dissertation "Measurement of oxygen clumped isotopes using mass spectrometry" by Stephan Räss.*

**RC 2**

The manuscript is too lengthy in its current form. It would benefit from improvements in language, as well as the removal of less important descriptions, elimination of repetitive content. Additionally, some figures (4-8), table (table 1) and a significant amount of texts could be moved to supplementary materials for better organization.

*Section 3.1 and 3.2 were moved to the appendix. The key findings were summarized at the end of Section 3.*

The manuscript mentions the mass spectrometer's ability to measure signal currents for masses 35 and 36 for O2, and associated error estimates are provided. Their statements at the beginning of introduction "we observed that our device is sensitive enough to measure the multiply-substituted oxygen isotopologues 17O18O and 18O18O in pure-oxygen gas, despite its low mass resolution and its use of 1011 Ω resistors on the corresponding cups" is a bit misleading. The isobaric intereference to 18O18O from 36Ar cannot be avoided but resolved in the mass spectrometer (for that a resolving power of ~1170 or better is required). This is because a sample is very unlikely be 100% Ar free (even after passing through GC) and the amount of Ar present in a sample from the background (even for pure O2) would also interfere 18O18O measurement.

*The sentence "We observed that our device is sensitive enough to measure the multiply-substituted oxygen isotopologues 17O18O and 18O18O in pure-oxygen gas, despite its low mass resolution and its use of 10^11 Ω resistors on the corresponding cups" was reformulated as "We observed that our device is sensitive enough to measure the multiply-substituted oxygen isotopologue 17O18O in pure-oxygen gas. We also detected a peak at $m/z = 36$ u e^-1, which corresponds primarily to 18O18O, though minor impurities may contribute small amounts of 36Ar."*

Although the primary focus is on monitoring background shifts rather than measuring 18O18O, 17O18O, the observed errors could stem from variable contributions from isobaric interferences such as 36Ar and 35Cl for 18O18O and 17O18O, respectively. In section 3, authors mentioned that they cannot distinguish 17O17O from 16O18O due to low resolving power of the mass spectrometer used. The mass resolving power is not enough to separate 36Ar from 18O18O, let's leave separating 17O17O from 16O18O. For separation of 17O17O from 16O18O, mass resolving power required is 8117. I would say separation of these two isotopologues is not possible even with a mass resolution of >8117 because of the intensity imbalance of the two isotopologues (16O18O signal is >27,000 times more than 17O17O, 10 % valley definition is practically inapplicable). Therefore, I suggest authors to make the statements a bit carefully. I Also suggest authors to discuss these somewhere in their manuscript.

*The sentence "Due to isobaric interferences of oxygen components, with our setup, it is not possible to distinguish the clumped isotope 17O17O (m/z = 34) from 16O18O." was replaced by "It should also be noted that, with our setup, it is not possible to detect 17O17O (m/z = 34). First, there is an isobaric interference with 16O18O; second, there is a large intensity difference between 17O17O and 16O18O, making the detection of the former isotopologue challenging, even for high mass-resolution instruments (Laskar, 2019)."*

Please shorten the conclusion section.

*Done.*

Detailed comments on the manuscript are provided in the annotated PDF.

*We clarified the statements on lines 13/14 and 399, and we have removed and/or shortened the other highlighted sections as suggested.*

**RC 3**

The manuscript is well written and highly detailed but is overly long and repetitive. The principles discussed are valuable and may have broader applicability to other measurements and instruments. Nevertheless, the manuscript would benefit considerably from substantial condensation and a more results-focused revision.

*The introduction and conclusion were shortened. Furthermore, parts of Section 3 were moved to the appendix.*

Abstract: The abstract should, in my view, begin by stating that minor isotopes are significantly affected by secondary electrons, which are dependent on bulk mass flow. It should then outline that the manuscript addresses the methodology for correcting this effect.

*The abstract now starts as follows: "Secondary electrons emitted from the surfaces of Faraday cups can significantly influence the peak shapes and intensities of minor isotopes. As the amount of these electrons depends on the source pressure, pressure baseline corrections have been proposed to mitigate these pressure-dependent background effects and reduce the apparent dependence of $\Delta_{47}$ on $\delta_{47}$ (non-linearity) observed in clumped-isotope studies of $CO_2$."*

Line 10: The term on-peak signals is introduced without prior explanation. Clarification is needed at this stage.

*We replaced "on-peak signals" by "signals on the peak top (on-peak signals)".*

Line 13: The phrase at least 120 intervals is ambiguous. How many intervals have actually been used for the calculation of standard deviations?

35/32: 1e-9: Table 4, 10 intervals

$\delta_{35}$: 0.2 %: Table 3, 120 intervals

$\Delta_{35}$: 0.5 %: Sect. 4.4.1, 120 intervals

36/32: 7e-9: Table 6, 120 intervals

$\delta_{36}$: 0.1 %: Table 6, 230 intervals

$\Delta_{36}$: 0.1 %: Table 6, 230 intervals

*For simplicity, we dropped 'at least' because the standard deviation usually increases with the number of intervals. Furthermore, we already use 'typically' at the beginning of the sentence. The same adjustment was made in the conclusion.*

The introduction can be shortened significantly.

*Done.*

3.1 The conclusion of this section is that secondary electron suppression is not able to remove the secondary electrons fully and background corrections are still necessary. Please write this.

*The sentence "Most importantly, Fig. 2 illustrates that applying a negative potential to the Faraday cups is insufficient to achieve m/z = 36 signals that are higher than the collector zero value, resulting in negative background-corrected signals." was replaced by "Most importantly, Fig. A2 illustrates that the use of electron suppressors alone is insufficient to produce m/z = 36 signals above the collector zero value, resulting in negative background-corrected signals that necessitate further background correction."*

Figure 9: Please adjust the Y-scales.

*Done.*

The conclusions could be more concise. I recommend summarising the key successes of the approach and providing the achieved standard errors, ensuring the focus remains on the core findings.

*Done.*